

# Anisotropic P-wave traveltime tomography implementing Thomsen's weak approximation in TOMO3D

Adrià Meléndez[1], Clara Estela Jiménez[1], Valentí Sallarès[1], César R. Ranero[2]

[1]Barcelona Center for Subsurface Imaging, Institut de Ciències del Mar (CSIC), Barcelona, 08003, Spain
[2]Barcelona Center for Subsurface Imaging, ICREA at Institut de Ciències del Mar (CSIC), Barcelona, 08003, Spain

*Correspondence to*: Adrià Meléndez (melendez@icm.csic.es)

**Abstract.** We present the implementation of Thomsen's weak anisotropy approximation for VTI media within TOMO3D, our code for 2-D and 3-D joint refraction and reflection traveltime tomographic inversion. In addition to the inversion of seismic P-wave velocity and reflector depth, the code can now retrieve models of the Thomsen's parameters $\delta$ and $\varepsilon$. Here we test this new implementation following four different strategies on a canonical synthetic experiment. First, we study the sensitivity of traveltimes to the presence of a 25% anomaly in each of the parameters. Next, we invert for two combinations of parameters, ($v$, $\delta$, $\varepsilon$) and ($v$, $\delta$, $v^\perp$), following two inversion strategies, simultaneous and sequential, and compare the results to study their performances and discuss their advantages and disadvantages. Simultaneous inversion is the preferred strategy and the parameter combination ($v$, $\delta$, $\varepsilon$) produces the best overall results. The only advantage of the parameter combination ($v$, $\delta$, $v^\perp$) is a better recovery of the magnitude of $v$. In each case we derive the fourth parameter from the equation relating $\varepsilon$, $v^\perp$ and $v$. Recovery of $v$, $\varepsilon$ and $v^\perp$ is satisfactory whereas $\delta$ proves to be impossible to recover even in the most favorable scenario. However, this does not hinder the recovery of the other parameters, and we show that it is still possible to obtain a rough approximation of $\delta$ distribution in the medium by sampling a reasonable range of homogeneous initial $\delta$ models and averaging the final $\delta$ models that are satisfactory in terms of data fit.

## 1 Introduction

An isotropic velocity field is the rare exception in the Earth subsurface. Anisotropy is a multiscale phenomenon, and its causes are diverse. In the crust, it can be produced by the preferred orientation of mineral grains or their crystal axes (Schulte-Pelkum and Mahan, 2014; Almqvist and Mainprice, 2017), the alignment of cracks and fracture networks and the presence of fluids (Crampin, 1981; Maultzsch et al., 2003; Yousef and Angus, 2016), or the bedding of layers much thinner than the wavelength used to explore them (Backus, 1962; Johnston and Christensen, 1995; Sayers, 2005). In the mantle, anisotropy is related to the alignment of olivine crystals due to mantle flow (Nicolas and Christensen, 1987; Montagner et al., 2007), aligned melt inclusions (Holtzman et al., 2003; Kendall et al., 2005), large-scale deformation (Vinnik et al., 1992; Vauchez et al., 2000), and preexisting lithospheric fabric (Kendall et al., 2006), among others. Anisotropy has proven an informative physical property in the understanding of the Earth's interior (Ismaïl and Mainprice, 1998; Long and Becker,



2010), most particularly in continental rifts (Eilon et al., 2016), mid-ocean ridges (Dunn et al., 2001), and subduction zones (Long and Silver, 2008).

The theory of anisotropic wave propagation has been described in several publications (Kraut, 1963; Babuska and Cara, 1991). Numerous formulations of varying complexity have been proposed to approximate anisotropy depending on its
magnitude and the symmetry conditions of the medium (Nye, 1957). Twenty-one elastic stiffness parameters define the most general anisotropic medium with the lowest symmetry conditions, whereas for the highest symmetry not equivalent to isotropy, only five parameters are needed (Almqvist and Mainprice, 2017). Regarding the strength of anisotropy, in view of the overall success of isotropic methods in studying the Earth's subsurface, and of the experimental evidence and sample measurements available, it is admitted that the anisotropy is generally weak (Thomsen, 1986). Specifically, anisotropy is
considered weak when Thomsen's parameters are much smaller than 1, i.e. for a ~20% or smaller velocity variation with angle. Precisely Thomsen (1986) presented the formulation for the transverse isotropy symmetry on weakly anisotropic media, which is the reference that we follow in this work. Thomsen's parameters are by far the most common and convenient combinations of stiffness tensor elements used in seismic anisotropy modelling (Tsvankin, 1996; Thomsen and Anderson, 2015). Applications of simpler approximations exist, as is the case of elliptical symmetry (Song et al., 1998; Giroux and
Gloaguen, 2012), as well as others that assume the most general anisotropic model (Zhou and Greenhalgh, 2008).

The objectives of this work are (1) presenting the anisotropic version of TOMO3D (Meléndez et al., 2015) for the study of VTI weakly-anisotropic media in terms of the Thomsen's parameters $\delta$ and $\varepsilon$ using P-wave arrival times, and (2) comparing several parametrizations and inversion strategies to identify those providing the most accurate results. The P-wave velocity,
$\delta$ and $\varepsilon$ models obtained from the modeling of field data would be useful and geologically informative by themselves, but they would also serve as initial models in anisotropic FWI. Moreover, the development of this code is motivated by the need to combine wide-angle and near-vertical traveltime picks in field data applications, as we plan to do with the trench-parallel 2-D profile in Sallarès et al. (2013), which is affected by a ~15% anisotropy judging from the mismatch in the interplate boundary locations obtained separately from near-vertical and wide-angle data.


In the following section, we describe the anisotropy formulation and the modifications implemented on our 3-D joint refraction and reflection traveltime tomography code TOMO3D to incorporate the inversion of Thomsen's parameters. Next, in the third section, we present the synthetic tests performed and their results, including accuracy and sensitivity analyses and synthetic inversions. In section four, these results are discussed and interpreted in terms of the ability of the code to retrieve
both the velocity field and Thomsen's anisotropy parameters. Finally, in the last section we summarize the main conclusions of this work.



## 2 Modelling anisotropy

The first part of this section is a general overview of the treatment of anisotropy within the field of seismic inversion, while the second one describes the implementation of Thomsen's weak anisotropy formulation in the 3-D joint refraction and reflection traveltime tormography code TOMO3D.

### 2.1 Anisotropy in seismic inversion methods

Anisotropy was first incorporated to seismic inversion methods in traveltime tomography with the development of the linearized perturbation theory (Cervený, 1982; Cervený and Jech, 1982; Jech and Psencik, 1989). Previously, the approach to deal with anisotropy was to approximately remove its estimated effect to then apply an isotropic method (e.g. McCann et al., 1989). Linearized perturbation theory was first implemented in anisotropic traveltime tomography by Chapman and Pratt (1992) and Pratt and Chapman (1992) assuming the weak anisotropy approximation, which allowed them to use isotropic ray tracing and approximate anisotropy effects as being caused by small perturbations of the isotropic system. The initial development of anisotropic ray tracing is attributed to Cervený (1972). Methods for anisotropic ray tracing and traveltime computation depend on the symmetry assumptions made regarding the medium. The most common of those is rotational symmetry around a vertical pole. This formulation is known as vertical transverse isotropy (VTI), also polar anisotropy (e.g. Rüger and Alkhalifah, 1996; Alkhalifah, 2002), and it is the simplest geologically applicable case: it reproduces the symmetry exhibited by minerals in sedimentary rocks, and that produced by parallel cracks or fine layering. Furthermore, it significantly simplifies the mathematical formulae since anisotropy is defined by only five parameters, which contributes to a greater computational efficiency. The generalization of VTI to a tilted symmetry axis is the so called tilted transverse isotropy (TTI). Some authors argue that it is not possible to distinguish TTI from VTI in real experimental cases without a priori information (Bakulin et al., 2009). Assuming the most general anisotropic media has also become rather usual, in particular with the improvement of computational resources, allowing for a more detailed and complex reconstruction of the subsurface physical properties (e.g. Zhou and Greenhalgh, 2005) although successful field data applications are yet to be achieved to the best of our knowledge. Regarding the inversion process, the main difficulty arises from the trade-off between velocity heterogeneity and anisotropy (e.g. Bezada et al., 2014). To the best of our knowledge, Stewart (1988) was the first to propose an inversion algorithm, specifically for the recovery of Thomsen's parameters in a weakly-anisotropic VTI medium. Other authors have produced inversion algorithms for different formulations such as azimuthal anisotropy (e.g. Eberhart-Phillips and Henderson, 2004; Dunn et al., 2005) or a 3-D TTI medium (e.g. Zhou and Greenhalgh, 2008). Concerning FWI, anisotropy in active data is typically modeled following Thomsen's parameters and the VTI and/or TTI approximation for the medium. The first anisotropic wave propagators appeared during the 80s and 90s (e.g. Helbig, 1983; Alkhalifah, 1998) and new improvements on this matter continue today (e.g. Fowler et al., 2010; Duveneck and Bakker, 2011). When performing anisotropic full waveform inversion (FWI), both 2-D and 3-D, some authors choose to invert only for the velocity field, fixing the initial anisotropy models throughout the inversion because it simplifies the process (e.g.



Prieux et al., 2011; Warner et al., 2013). However, other works have explored the feasibility of multiparameter inversions,
that is, using different combinations of velocity and anisotropy parameters and of inversion strategies (e.g. Gholami et al., 2013a,b; Alkhalifah and Plessix, 2014).

## 2.2 Anisotropy in TOMO3D: Thomsen's weak anisotropy formulation

We adapted TOMO3D (Meléndez et al., 2015) to perform anisotropic ray tracing and traveltime calculations, as well as inversion of Thomsen's parameters for P-wave data. In TOMO3D, the forward problem solver is parallelized to
simultaneously trace rays for multiple sources and receivers, and it uses an hybrid ray tracing algorithm that combines the graph or shortest path method (Moser et al, 1991) and the bending refinement method (Moser, 1992). The inverse problem is solved sequentially using the LSQR algorithm (Paige and Saunders, 1982). Velocity models are discretized as 3-D orthogonal and vertically-sheared grids that can account for topography and/or bathymetry. Apart from first-arrival traveltimes, the code allows for the inversion of reflection traveltimes to obtain the geometry of major geological boundaries
associated to impedance contrasts that produce strong seismic energy reflections in the data recordings. Such reflecting interfaces are modeled as 2-D grids independent of the velocity grid. The code is also prepared to extract information from the water-layer multiple of refracted and reflected seismic phases (Meléndez et al., 2013). A detailed description of the code can be found in Meléndez (2014).

Our anisotropy formulation is based on Thomsen (1986) and specifically in the following weakly-anisotropic velocity equation for the P-wave velocity:

$$v^a(v, \delta, \varepsilon, \theta) = v \cdot \left(1 + \delta \cdot sin^2(\theta) \cdot cos^2(\theta) + \varepsilon \cdot sin^4(\theta)\right) \tag{1}$$

where $v^a$ is the anisotropic velocity, $v$ is the velocity along the symmetry axis ($\alpha_0$ in Thomsen (1986)), $\theta$ is the angle with respect to the symmetry axis, and $\delta$ and $\varepsilon$ are Thomsen's parameters controlling the anisotropic P-wave propagation. Studying the cases of $\theta=0$ and $\theta=\pi/2$ the meaning of $\varepsilon$ becomes clear: it is the relative difference between the velocities along and across the symmetry axis, that we refer to as parallel and perpendicular velocities, respectively.

$$v^a(v, \theta = 0) = v$$


$$v^a\left(v, \theta = \frac{\pi}{2}\right) = v \cdot (1 + \varepsilon) \equiv v^\perp$$

$$\varepsilon = (v^\perp - v)/v \tag{2}$$





According to Thomsen (1986) $\delta$ is related to the near-vertical anisotropic response but its meaning is far from intuitive.
However, a mathematical relationship between $\delta$, $v$, and the normal move-out velocity ($V_{NMO}$) exists. $V_{NMO}$ models are built
as part of the normal move-out correction in seismic reflection data processing. At best, our traveltime tomographic method
would be able to produce approximations of the actual $V_{NMO}$ models. Furthermore, such approximations would only be
meaningful, if ever, when derived from travel times of a seismic reflection data set, for which the normal move-out
correction and thus the $V_{NMO}$ are defined. Of course, in such a case, actual $V_{NMO}$ models would be obtained from the normal
move-out correction, and therefore $\delta$ could be calculated provided that a $v$ model is available, for instance from our
traveltime tomography. Thus, we only consider Eq. (2), and we implemented two parametrizations of the medium: ($v$, $\delta$, $\varepsilon$)
and ($v$, $\delta$, $v^\perp$). From here on, for simplicity, we will refer to them as P[$\varepsilon$] and P[$v^\perp$] respectively.

The linearized inverse problem matrix equation including anisotropy parameters is as follows

$$
\begin{pmatrix} \Delta t^0 \\ \Delta t^1 \\ 0 \\ 0 \\ 0 \\ 0 \\ 0 \\ 0 \\ 0 \\ 0 \\ 0 \\ 0 \\ 0 \\ 0 \\ 0 \\ 0 \\ 0 \\ 0 \\ 0 \end{pmatrix} =
\begin{pmatrix}
\boldsymbol{G}^{u0} & 0 & \boldsymbol{G}^{\delta 0} & \boldsymbol{G}^{\varepsilon 0} \\
\boldsymbol{G}^{u1} & w\boldsymbol{G}^{z} & \boldsymbol{G}^{\delta 1} & \boldsymbol{G}^{\varepsilon 1} \\
\lambda_u \boldsymbol{L}^{uX} & 0 & 0 & 0 \\
\lambda_u \boldsymbol{L}^{uY} & 0 & 0 & 0 \\
\lambda_u \boldsymbol{L}^{uZ} & 0 & 0 & 0 \\
0 & w\lambda_z \boldsymbol{L}^{zX} & 0 & 0 \\
0 & w\lambda_z \boldsymbol{L}^{zY} & 0 & 0 \\
0 & 0 & \lambda_\delta \boldsymbol{L}^{\delta X} & 0 \\
0 & 0 & \lambda_\delta \boldsymbol{L}^{\delta Y} & 0 \\
0 & 0 & \lambda_\delta \boldsymbol{L}^{\delta Z} & 0 \\
0 & 0 & 0 & \lambda_\varepsilon \boldsymbol{L}^{\varepsilon X} \\
0 & 0 & 0 & \lambda_\varepsilon \boldsymbol{L}^{\varepsilon Y} \\
0 & 0 & 0 & \lambda_\varepsilon \boldsymbol{L}^{\varepsilon Z} \\
\alpha_u \boldsymbol{D}^{u} & 0 & 0 & 0 \\
0 & w\alpha_z \boldsymbol{D}^{z} & 0 & 0 \\
0 & 0 & \alpha_\delta \boldsymbol{D}^{\delta} & 0 \\
0 & 0 & 0 & \alpha_\varepsilon \boldsymbol{D}^{\varepsilon}
\end{pmatrix}
\begin{pmatrix} \Delta u \\ \frac{1}{w}\Delta z \\ \Delta \delta \\ \Delta \varepsilon \end{pmatrix}
\tag{3}
$$

Smoothing ($\boldsymbol{L}$) and damping ($\boldsymbol{D}$) constraints for $\delta$ and $\varepsilon$ parameters follow the same formulation described in Meléndez
(2014) for velocity parameters. The kernels ($\boldsymbol{G}$) have been modified to account for anisotropy. The linearized and discretized
equation that relates the traveltime residual of the $n$-th refracted pick to changes in the model parameters is written as

$$
\Delta t_n^0 = \sum_i^I \sum_{m=1}^8 r_m^u \cdot \frac{\partial t}{\partial u^a} \cdot \frac{\partial u^a}{\partial u} \cdot \Delta u_{im} + \sum_j^J \sum_{m=1}^8 r_m^\delta \cdot \frac{\partial t}{\partial u^a} \cdot \frac{\partial u^a}{\partial \delta} \cdot \Delta \delta_{jm} + \sum_k^K \sum_{m=1}^8 r_m^\varepsilon \cdot \frac{\partial t}{\partial u^a} \cdot \frac{\partial u^a}{\partial \varepsilon} \cdot \Delta \varepsilon_{km}
\tag{4}
$$

In each of the three terms, the first summation corresponds to the model cells that are illuminated by the $n$-th ray path. A cell
is considered illuminated if it contains a ray path segment. The second summation is over the eight nodes of each of those




cells. In the third term, epsilon is replaced by $v^\perp$ when using this alternative parametrization. $\Delta t_n^0$ is the $n$-th traveltime residual for a refracted pick, $u^a = 1/v^a$ is the anisotropic slowness, $u = 1/v$ is the along-axis slowness, $\Delta u$, $\Delta \delta$, and $\Delta \varepsilon$ (or $\Delta v^\perp$) are the parameter perturbations for each node of each illuminated cell, and $r$ factors are weights that distribute the kernel value for each cell among its eight nodes according to the trilinear interpolation used to define the four fields ($u$, $\delta$, $\varepsilon$, and $v^\perp$) based on the values at the mesh nodes.


In order to build the kernel matrices we need to compute two partial derivatives for each model parameter. The first order partial derivative of traveltime with respect to the anisotropic slowness is the ray path segment $s_i$ within each cell that is covered in a given traveltime at a given slowness

$$t = \sum_{i=1}^{N} u_i^a \cdot s_i \tag{5}$$

$$\frac{\partial t}{\partial u_i^a} = s_i \tag{6}$$

From Eq. (1) the first order partial derivatives of the anisotropic slowness with respect to the model parameters ($u$, $\delta$, $\varepsilon$, and
$v^\perp$) are as follows

$$\frac{\partial u^a}{\partial u} = \frac{1}{1 + \delta \cdot \sin^2(\theta) \cdot \cos^2(\theta) + \varepsilon \cdot \sin^4(\theta)} \tag{7}$$

$$\frac{\partial u^a}{\partial \delta} = \frac{-u \cdot \sin^2(\theta) \cdot \cos^2(\theta)}{\left(1 + \delta \cdot \sin^2(\theta) \cdot \cos^2(\theta) + \varepsilon \cdot \sin^4(\theta)\right)^2} \tag{8}$$


$$\frac{\partial u^a}{\partial \varepsilon} = \frac{-u \cdot \sin^4(\theta)}{\left(1 + \delta \cdot \sin^2(\theta) \cdot \cos^2(\theta) + \varepsilon \cdot \sin^4(\theta)\right)^2} \tag{9}$$

$$\frac{\partial u^a}{\partial v^\perp} = \frac{-(u)^2 \cdot \sin^4(\theta)}{\left(1 + \delta \cdot \sin^2(\theta) \cdot \cos^2(\theta) + \varepsilon \cdot \sin^4(\theta)\right)^2} \tag{10}$$

If the $n$-th pick corresponds to a reflected ray, then its traveltime residual is related to changes in model parameters by the following equations

$$\Delta t_n^1 = \Delta t_n^0 + \sum_{m=1}^{4} r_m^z \cdot \frac{\partial t}{\partial z} \cdot \Delta z_m \tag{11}$$

with





$$\frac{\partial t}{\partial z} = \sum_{m=1}^{4} r_m^z \cdot \left( u^a(\theta_i) + u^a(\theta_r) \right) \cdot cos(\eta) \cdot cos(\xi) \qquad (12)$$

where the additional term corresponds to the depth kernel which has been modified to account for anisotropy from its isotropic version as derived by Bishop et al (1985). $u^a(\theta_i)$ and $u^a(\theta_r)$ are the anisotropic slownesses at the reflecting point on the interface for the incident and reflected rays respectively, $\eta$ is the angle of the interface with respect to the horizontal, and

$\xi$ is the incidence angle with respect to the interface normal vector. For simplicity, in this work we focus the analysis on first arrival inversion so we do not use reflection picks.

## 3. Synthetic tests

We have performed a number of tests using canonical synthetic models made of an anomaly centered in a uniform background with two main objectives: (1) checking that the newly implemented anisotropic traveltime tomography method

works properly, and (2) providing a quantitative measure of the potential recovery of anisotropy based on P-wave traveltimes alone. To test this we first calibrated the code by comparing the synthetic data that it generates to analytically calculated data. Second, we run a sensitivity test to assess the effect that a variation in each model parameter has in the synthetic traveltimes. Finally, we performed a number of synthetic inversion tests considering both possible parametrizations and inversion strategies.


The models in all these tests are cubes of 5-km-long edge. The background model of all four parameters is set to a constant value, i.e. $v$, $\delta$, $\varepsilon$ and $v^{\perp}$ background models are homogeneous. Note that the z axis positive direction points downwards. Grid spacing is 0.125 km for the four parameters in all three dimensions so that differences in model discretization do not influence the test results. The volume of the anomaly is determined by the $3\sigma$ region of a 3-D Gaussian function centered in

the cube setting $3\sigma = 0.5$ km. The values of the grid nodes within this volume are homogeneously increased by 25% resulting in a discretized representation of a spherical anomalous body.

### 3.1 Accuracy

Alternately for each of the four parameters, the said anomaly was added at the center of the cube representing a 25% increase on the background value, while the models for the rest of parameters remained homogeneous. Table 1 summarizes the

background and anomaly values for all parameters.

The acquisition configuration consists of 482 diametrically-opposed source–receiver pairs. Each receiver records exclusively the first arrival traveltime from its corresponding source for a total of 482 traveltimes (Fig. 1). Sources and receivers are located at 2.5-km of the center of the model, at the locus defined by the surface of the sphere inscribed in the cube, and

placed at the crossing points of 32 meridians with 15 parallels, and at each pole.

In these four cases, we compared the synthetic traveltimes obtained with our code to the analytic solution. The comparison along two selected meridians at 0 rad and $\pi/4$ rad azimuths is displayed in Fig. 2. Table 2 contains the mean traveltime misfits relative to the analytic traveltimes in percentage and their respective mean deviations, for each of these four tests and
along the two selected meridians, as well as the overall values.

### 3.2 Sensitivity

For the calculation of sensitivities, we used the same data, that is, the same models and acquisition geometry, as for the accuracy test. For instance, when computing the sensitivity of velocity, we add the 25% spherical anomaly to the velocity background model, while the models for the other parameters remain homogeneous.


We define sensitivity as the difference between the first arrival traveltimes with and without the anomaly. Figure 3 shows synthetic and analytic sensitivities for the two selected meridians and the four parameters. Note that the analytic response does not change between meridians, i.e. given the symmetry of the models and of the anisotropic formulation, sensitivity is independent of the azimuth angle.


Comparison of Fig. 3a,b with the values in Table 2 and Fig. 2 indicates that the forward calculation of traveltimes is accurate enough with respect to the traveltime residuals expected for the selected anomalies. Sensitivity is at least 5 times and up to two orders of magnitude greater than traveltime accuracy depending on the parameters, with the exception of angles for which sensitivity tends to zero.


$v$ sensitivity is the highest for all angles, 4.5% to 5% (Fig. 3). In the direction perpendicular to the symmetry axis, $v^\perp$ sensitivity is virtually identical to $v$ sensitivity, but as we move away from this direction sensitivity decreases to 0% at the direction of the symmetry axis, as expected from Eq. (1). $\varepsilon$ sensitivity logically follows the same angular dependence as $v^\perp$ but its magnitude is as much as 5 times smaller than that of $v$, ~0.8% at its maxima. Finally, $\delta$ sensitivity is, at its maxima,
one order of magnitude smaller than $v$ sensitivity, that is around 0.25%. These sensitivity results indicate that we can generally expect a better recovery of $v$ than of $\varepsilon$ and $v^\perp$, and that retrieving $\delta$ might prove complicated. Keep in mind that these sensitivities are produced by an anomaly that represents a 25% increase with respect to the background value.

The differences in synthetic sensitivities between meridians arise from the discretization of the model space in a Cartesian
system of coordinates. Such approximation inevitably defines privileged directions for ray tracing, and consequently produces differences in synthetic traveltimes. Regarding the mismatch between synthetic and analytic sensitivities (Fig. 3), it occurs because the discretization used cannot represent the surface of a perfect sphere. These effects are most notable in the $v$ sensitivity, and to a lesser extent in $v^\perp$, precisely because these are the most sensitive parameters, and thus the errors in the

representation of a sphere and the existence of privileged directions have a much larger influence on the calculated

traveltimes. Figure S1 shows how refining the *v* model generates a much more accurate sensitivity pattern, and reduces the

relative traveltime misfit. In a real case study one can always refine the grid spacing of a particular parameter to achieve

better accuracy, but here we wish to test the performance of the code in the modeling and recovery of each parameter under

the same conditions, i.e. equivalent anomalies and identical model discretization.

### 3.3. Inversion results

For the inversion tests, we considered a synthetic medium defined by the anomaly models of all four parameters. Here we

refer to these models as target models, and the goal of the inversion is to retrieve the heterogeneity in each of them. These

tests are conducted for the two parametrizations of the anisotropic medium described in section 2: P[$\varepsilon$] and P[$v^\perp$]. Note that,

in order to perform the inversion tests on equivalent cases for both parametrizations, the heterogeneity in $v^\perp$ is calculated

with Eq. (2) considering the 25% anomalies in *v* and $\varepsilon$, which yields a ~29.3% anomaly in $v^\perp$ (Table 3). If not indicated

otherwise, we use background models as initial models. Finally, we study the potential recovery of $\delta$ because inverting this

particular parameter proves notoriously difficult due to its low sensitivity (Fig. 3).

In this case, the acquisition geometry is made of 114 sources each with 113 receivers (Fig. 1). Again, sources and receivers

are located at the surface defined by the sphere inscribed in the cube, but now all receivers record all sources, except for the

one coinciding in location.

For both parametrizations, we compared two inversion strategies: simultaneously inverting for all parameters and a two-step

sequential inversion. First, in Figs 4 and 5 we show the best results for the simultaneous inversion strategy. For each

parametrization we derived the fourth parameter applying Eq. (2).


Table 4 shows several statistical measures to quantify the quality of these inversion results. As a measure of data fit

improvement, we provide the RMS of traveltime residuals for the first and last iterations. As a measure of model recovery or

fit, for each parameter, we calculate the mean relative misfits for the background area between the inverted model and either

the target or the initial one, as they are identical in this area, as well as for the anomaly area comparing the inverted model to

both the target and the initial models. In the case of a perfect recovery, mean relative misfit for the background area would

be 0%, whereas for the anomaly area, it would be 0% when using the target model as a reference, and 25% (~29.3% for $v^\perp$)

when comparing to the initial model.

In an attempt to improve the recovery of $\delta$, we repeated these two tests for different values of the smoothing constraints, but

it proved impossible. Correlation lengths tested for all four parameters include 0.25 km (twice the grid spacing), 0.5 km, and

1 km. The weights of the smoothing submatrices for each parameter, $\lambda$ in Eq. (3), were varied between 1 and 100, with





intermediate values of 2, 5, 10, 20, 30, and 60. For successful inversions, very similar results for the other parameters were obtained regardless of the final $\delta$ model. In other words, the low sensitivity of $\delta$ makes it impossible to recover it from traveltime data, but for this same reason it has little or no influence on the recovery of $v$ and $\varepsilon$ or $v^{\perp}$.


The two-step sequential inversion strategy was also tested for both parametrizations P[$\varepsilon$] and P[$v^{\perp}$]. For the first step, we tested two options : (a) inverting for $v$ while fixing $\delta$ and $\varepsilon$ or $v^{\perp}$ and (b) fixing only $\delta$. In the second step we used the inverted models from step 1 as initial models and tested three options: (c) inverting for all three parameters, (d) fixing only $\delta$, when following option (a) in step 1, and (e) fixing $v$ and/or $\varepsilon$ or $v^{\perp}$, when following option (b) in step 1. Again smoothing

constraints were varied for similar correlation lengths and submatrix weights as detailed for the simultaneous inversion case.

In the case of P[$\varepsilon$], the best result (Fig. 6) was obtained inverting for $v$ and $\varepsilon$ while fixing $\delta$ in the first step, and fixing only $\varepsilon$ in the second step. As for P[$v^{\perp}$], the best combination for the two-step inversion (Fig. 7) was fixing only $\delta$ in step 1, and inverting for the three parameters in step 2. Tables 5 and 6 summarize the statistical quantification of data and model fit for

each parametrization.

### 3.4 Modelling $\delta$

Observing that good results for $v$ and $\varepsilon$ or $v^{\perp}$ are achieved regardless of the result in $\delta$, and knowing that the sensitivity of $\delta$ is notably smaller than that of the other parameters, we explored a strategy to have an estimate of this parameter. First, as a reference, we considered an unrealistically optimal scenario in which the real $v$ and $\varepsilon$ or $v^{\perp}$ models are known to us. Figs 8

and 9 show the resulting $\delta$ achieved by repeating inversions in Figs 4 and 5 with $v$ and $\varepsilon$ or $v^{\perp}$ target models as initial models. Table 7 summarizes the traveltime residuals RMS and the mean relative misfits of each parameter for these inversions. Again, these two tests were repeated for ranges of smoothing constraints in all four parameters, as described for the cases in Figs 4 and 5. Table 7 and Figs 8 and 9 correspond to the best results obtained, which indicate that the recovery of $\delta$ is, at best, extremely complicated due to the limited sensitivity of traveltime data to changes in this parameter.


Next, we decided to try neglecting $\delta$ in Eq. (1), and we repeated a number of inversions, such as the ones displayed in Figs 4 and 5, following the equation

$$v^a(v,\varepsilon,\theta)=v \cdot \left(1 + \varepsilon \cdot sin^4(\theta)\right) \tag{13}$$


The purpose of these tests was checking whether it was possible to invert $v$ and $\varepsilon$ or $v^{\perp}$ with data generated following Eq. (1) using the approximation in Eq. (13) given that the influence of $\delta$ on the results for other parameters is rather small, that $\delta$ cannot be accurately retrieved from traveltime alone, and that it has the smallest sensitivity. To do so, an homogeneous

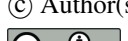


model of $\delta$=0 was fixed throughout the inversions. These tests were unsuccessful, with noticeably poorer results than when considering a dependence on $\delta$ (Table 8). However, they were useful in proving that even if a detailed $\delta$ model is not necessary to successfully retrieve the other parameters, at least a rough approximation of the $\delta$ field is needed to recover the other parameters, e.g. the background $\delta$ model that we used as initial model in inversions displayed in Figs 4 and 5.

Finally, we tested whether it would be possible to obtain at least this rough approximation of $\delta$ in the medium, valid in the sense that it allows for the successful recovery of the rest of parameters. Once again we repeated inversions from Figs 4 and 5 (initial $\delta = 0.16$), now using different homogeneous initial models for $\delta$ within a range of possible values from 0.1 to 0.24. Table 9 contains the initial and final RMS of traveltime residuals, as well as $\delta$ mean values for the inverted model along with the corresponding mean deviations. It is straightforward to note that, for a central subrange of the tested initial $\delta$ values, final RMS values are an order of magnitude smaller, a few tenths of millisecond compared to the few milliseconds outside this subrange. Specifically, very similar results to those in Figs 4 and 5 in terms of traveltime residuals RMS are produced by initial $\delta$ values between 0.13 and 0.22 for P[$\varepsilon$], and between 0.12 and 0.22 for P[$v^\perp$]. The narrowing of the initial $\delta$ distribution to a smaller subrange of mean $\delta$ values for the inverted models is indicative of a good general convergence trend.

The rough estimate of the $\delta$ field could be built, for instance, as the average of the mean $\delta$ values for the inverted models in the central subranges defined by the change in magnitude of the final RMS of traveltime residuals. One final inversion could be run using a homogeneous initial $\delta$ model with this average value, with the additional option of fixing it and inverting only for the other two parameters. As mentioned in subsection 2.2, potentially more detailed initial $\delta$ models could be obtained from the normal move-out correction of near-vertical reflection seismic data.

## 4. Discussion

We have tested two parametrizations of the VTI anisotropic media, P[$\varepsilon$] ($v$, $\delta$, $\varepsilon$) and P[$v^\perp$] ($v$, $\delta$, $v^\perp$), and two inversion strategies, the simultaneously inverting for all three parameters and a two-step sequential process fixing some of the parameters in each step. We consider three criteria for evaluating and comparing the quality of the inversion results obtained following the four possible combinations of strategies and parametrizations: visual inspection of the results, as well as traveltime data and model fits.

For the simultaneous inversion, both parametrizations were able to produce acceptable final results (Figs 4 and 5). According to our tests, P[$\varepsilon$] provides the best outcome, specifically because data and model fits (Table 4) are better for this option but, more importantly, because of the difference in the quality of the recovery of $\varepsilon$. Also, the recovery of $\delta$, even though it is far from acceptable, is significantly better in the case of P[$\varepsilon$]. However, visual comparison of the recovery of $v$ indicates that





P[$v^\perp$] yields a slightly better result regarding the magnitude of the anomaly for this parameter, as is particularly evident at the center of the anomaly. The boundaries of the anomalies for both velocities are still better retrieved with P[$\varepsilon$].

In general, sequential inversion is a more complex process that requires more human intervention and fine tuning in each step. In addition, fixing some of the parameters in the first step may result in the inverted parameters artificially accounting

for part of the data misfit that is actually related to the fixed ones. This can lead convergence into a local minimum, and it might be impossible to correct this tendency in the second step. For this inversion strategy, it is also P[$\varepsilon$] that produces the best results (Figs 6 and 7). Data fit and the model fit of $\varepsilon$ are slightly better than for the simultaneous inversion of this parametrization, whereas model fits and the visual aspect of both velocities are almost identical to those obtained by simultaneously inverting all three parameters (Tables 4 and 5). Visually, it is difficult to decide whether the recovery of $\varepsilon$ is

better or not than for the simultaneous inversion (Figs 4 and 6). As for $\delta$, recovery is unsuccessful and artifacts appear in the background area of the model but, according to both its model fit and its visual aspect, it is notably better than for the simultaneous inversion. As in the case of simultaneous inversion, the only advantage of using P[$v^\perp$] instead of P[$\varepsilon$] is that it yields a better recovery of the anomaly magnitude of $v$ (Tables 4 and 6). The results for both velocities are virtually identical to those obtained by simultaneous inversion of this parametrization (Figs 5 and 7). $\delta$ and $\varepsilon$ are not properly retrieved, but the

results are significantly better than for the simultaneous inversion of this same parametrization.

For both inversion strategies, results are always better for P[$\varepsilon$] than for P[$v^\perp$]. This behavior can be attributed to crosstalk between $v$ and $v^\perp$ as indicated by their sensitivities (Fig. 3). Indeed, for a moderately wide range of angles, roughly within $\pi/2 \pm \pi/4$ and $3\pi/2 \pm \pi/4$, both sensitivity patterns are of the same order of magnitude which can result in a substantial trade-

off for data within those ranges. Contrarily, the sensitivity patterns of the parameters used in P[$\varepsilon$] do not interfere as much with each other given their differences in magnitude.

$\delta$ has been shown to be by far the most complicated parameter to retrieve because of the low sensitivity of traveltime to its variation (Fig. 3). Even when excellent $v$ and $\varepsilon$ or $v^\perp$ models are available, i.e. the target models for these parameters in our

synthetic tests, the recovery of $\delta$ is limited at best (Figs 8 and 9 and Table 8). However, and for the same reason, poor recoveries of $\delta$ do not affect the recovery of the other two parameters, meaning that a detailed $\delta$ model is not necessary to satisfactorily retrieve $v$ and $\varepsilon$ or $v^\perp$ (Figs 4-9). Still, our inversion tests also proved that neglecting $\delta$ in Eqs (1) and (3) is not an option, the accuracy in the recovery of the other parameters resulting severely affected (Table 8). Thus, even if a detailed inversion of $\delta$ is, at the very least, hard to achieve, and it is not needed for a successful result in the other parameters, some

sort of simple, even homogeneous, initial $\delta$ model with a value or values about the average $\delta$ in the medium, is necessary for a good recovery of the other parameters.





We showed that given some a priori information on the range of possible $\delta$ values in the medium, it should be possible to create the necessary initial $\delta$ model. In order to illustrate this, we chose a range of $\delta$ values for the initial model and rerun the

inversions in Figs 4 and 5. The results indicate that for any homogeneous initial $\delta$ model in a certain subrange close to the actual average $\delta$ value of the medium, the results for $v$ and $\varepsilon$ or $v^\perp$ are satisfactory and virtually identical to those of the original inversions (Table 9). This subrange is easily defined by looking at the final RMS of traveltime residuals, which experiences a notorious change of an order of magnitude. Any model within this subrange works similarly well as initial $\delta$ model. Alternatively, a possibly more robust selection of the constant value for an homogeneous initial $\delta$ model might be the

mean (or also the median or the mode) of the mean $\delta$ values for the inverted models in this subrange. It is worth noting that, whereas for the purpose of this work we used the same discretization for all parameters, in a real case study it would probably be recommendable to use a coarser discretization for $\delta$ than for the other parameters, and in general a finer discretization for the more sensitive parameters (Fig. S1).

## 5 Conclusions

We have successfully implemented and tested a new anisotropic traveltime tomography code. For this implementation we had to modify both the forward problem and the inversion algorithms of the TOMO3D code (Meléndez et al., 2015). The forward problem was adapted to compute the velocities observed by rays considering Eq. (1) for the weak anisotropy formulation in Thomsen (1986). The inversion solver was extended to include the $\delta$, $\varepsilon$ and $v^\perp$ kernels in the linearized forward problem matrix equation, as well as smoothing and damping matrices for these parameters defined following the

same scheme as for velocity in the isotropic code (Eqs 3-12).

Regarding the synthetic tests, after checking the proper performance of the code by comparing with analytic solutions (Fig. 2), we determined the sensitivity of traveltime data to changes in each of the parameters defining anisotropy in the medium of interest ($v$, $\delta$, $\varepsilon$, $v^\perp$) (Fig. 3). Next, we performed inversion tests to compare two possible media parametrization, P[$\varepsilon$] and

P[$v^\perp$], and two possible inversion strategies, simultaneous and sequential. According to our tests, both parametrizations have their strengths: P[$\varepsilon$] produces the best overall result in the sense that all parameters are acceptably recovered and trade-off between parameters is lower, but P[$v^\perp$] yields the best result for the magnitude of the anomaly in $v$. Regarding the inversion strategy, simultaneous inversion is more straightforward and involves less human intervention, and given that both strategies yield similar results, it would be our first choice. Sequential inversion is always a more complex process that can be shown

to work in a synthetic case because the target models are available, but in field data applications the complexity would most likely be unmanageable.

An acceptable recovery of $\delta$ turned out to be impossible due to the small sensitivity of traveltimes to this parameter, but we verified that it cannot simply be neglected in the equations. Whereas the recovery of the other parameters is not significantly



affected by that of $\delta$, a rough estimate of the average $\delta$ value in the medium is necessary and sufficient to generate an homogeneous initial model that allows for satisfactory inversion results in the other parameters. We also proved that it is possible to obtain it, provided that some a priori knowledge on $\delta$ values in the medium is available to define a range of plausible values, such as field or laboratory measurements.

**Code availability**

The anisotropic version of TOMO3D will be made available for academic purposes only on our group website. Currently a copy of the code can be obtained by sending an e-mail to the corresponding author.

**Author contribution**

The formulation of the overarching research goals of this work is a product of discussion among the four co-authors. AM was in charge of software development and data curation, analysis, visualization, and validation. AM also prepared the
manuscript. The methodology for the synthetic tests was designed by AM, CEJ, and VS. CR was responsible for funding acquisition.

**Competing interests**

The authors declare that they have no conflict of interest.

**Acknowledgements**

AM and CLJ are funded by Respol through the SOUND collaboration project with CSIC, and the work in this paper was conducted at the Grup de Recerca de la Generalitat de Catalunya 2009SGR146: Barcelona Center for Subsurface Imaging (B-CSI). We thank all our fellows at the B-CSI for their contribution to this work. We also wish to thank Dr. Marko Riedel for his constructive comments at the 18th edition of the biennial International Symposium on Deep Seismic Profiling of the Continents and their Margins (SEISMIX 2018).

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

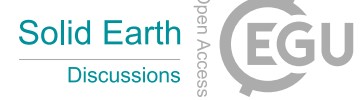



**Tables**

|  | $v$ (km/s) | $\delta$ | $\varepsilon$ | $v^{\perp}$ (km/s) |
|---|---|---|---|---|
| Background value | 2 | 0.16 | 0.16 | 2.32 |
| Anomaly value | 2.5 | 0.2 | 0.2 | 2.9 |

**Table 1: For accuracy and sensitivity tests, background and anomaly values of all four parameters for the initial/background model and the target models.**

| Anomaly in | $V$ | | $\delta$ | | $\varepsilon$ | | $v^{\perp}$ | |
|---|---|---|---|---|---|---|---|---|
| Azimuth | 0 rad | π/4 rad | 0 rad | π/4 rad | 0 rad | π/4 rad | 0 rad | π/4 rad |
| Mean relative misfit ± mean deviation (%) | 0.8 ± 0.1 | 0.6 ± 0.1 | 0.01 ± 0.01 | 0.01 ± 0.01 | 0.04 ± 0.05 | 0.03 ± 0.04 | 0.4 ± 0.3 | 0.4 ± 0.3 |
| | Overall values | | | | | | | |
| | 0.7 ± 0.1 | | 0.017 ± 0.009 | | 0.04 ± 0.03 | | 0.4 ± 0.2 | |

**Table 2: Mean relative traveltime misfits in percentage and their mean deviations for the two selected meridians in Fig. 2, and for the entire set of 482 source – receiver pairs. Compared to Fig. 3a,b, these average traveltime misfit values indicate that the code is sufficiently accurate to model the traveltime residuals arising from the inclusion of the selected anomalies.**

|  | $v$ (km/s) | $\delta$ | $\varepsilon$ | $v^{\perp}$ (km/s) |
|---|---|---|---|---|
| Background value | 2 | 0.16 | 0.16 | 2.32 |
| Anomaly value | 2.5 | 0.2 | 0.2 | 3 |

**Table 3: For inversion tests, background and anomaly values of all four parameters for the initial/background model and for the anomaly in the target model. The model is a cube of edge 5 km. The anomaly is a discretized sphere of 1 km in diameter at the center of the cube.**





|  | Residuals RMS (ms) | Mean relative misfits (%) | | | | | | | | | | | |
|---|---|---|---|---|---|---|---|---|---|---|---|---|---|
|  |  | $v$ | | | $\delta$ | | | $\varepsilon$ | | | $v^\perp$ | | |
|  |  | BG | AI | AT | BG | AI | AT | BG | AI | AT | BG | AI | AT |
| P[$\varepsilon$] | 30 – 0.4 | 0.5 | 21.0 | 3.3 | 4.8 | 22.3 | 15.2 | 1.6 | 11.1 | 11.2 | 0.5 | 22.8 | 5.0 |
| P[$v^\perp$] | 30 – 0.5 | 0.8 | 25.9 | 1.9 | 5.0 | 20.8 | 29.1 | 5.8 | 26.7 | 41.0 | 0.6 | 21.3 | 6.2 |

**Table 4: Quantification of the quality of recovery in terms of data and model fit for the simultaneous inversion of both parametrizations (Figs 4 and 5). Initial and final RMS values for traveltime residuals to quantify data fit. For each parameter, as a measure of model fit, we computed the mean relative misfits between the background areas of the inverted and target models (BG), the anomaly areas of the inverted and initial models (AI), and the anomaly areas of the inverted and target models (AT). Since the initial model is equal to the target model in the background area, it is not necessary to calculate the misfit between inverted and initial models for this area. The ideal misfit value for BG is 0%. AI and AT indicate the resemblance between true and recovered anomalies, and their ideal values are 0% and 25% (~29.3% in the case of $v^\perp$) respectively. Recovery is consistent with sensitivity (Fig. 3): $v$ and $v^\perp$ are well retrieved, $\varepsilon$ is only partially recovered with P[$\varepsilon$], whilst inversion of $\delta$ is unsuccessful in both cases.**

| P[$\varepsilon$] | Residuals RMS (ms) | Mean relative misfits (%) | | | | | | | | | | | |
|---|---|---|---|---|---|---|---|---|---|---|---|---|---|
|  |  | $v$ | | | $\delta$ | | | $\varepsilon$ | | | $v^\perp$ | | |
|  |  | BG | Ai | At | BG | Ai | At | BG | Ai | At | BG | Ai | At |
| Step 1 | 30 – 0.5 | 0.5 | 21.4 | 2.9 | - | - | - | 1.3 | 8.2 | 13.5 | 0.5 | 22.8 | 5.1 |
| Step 2 | 0.5 – 0.3 | 0.5 | 21.4 | 3.0 | 0.9 | 5.3 | 19.4 | - | - | - | 0.5 | 22.8 | 5.1 |

**Table 5: Same as Table 4 but for the two-step sequential inversion of P[$\varepsilon$] (Fig. 6). Recoveries in terms of model fit are virtually identical to those for the simultaneous inversion of this parametrization, with $\varepsilon$ recovery being just slightly better. Final data fit is also better than the one achieved by simultaneous inversion of P[$\varepsilon$].**

| P[$v^\perp$] | Residuals RMS (ms) | Mean relative misfits (%) | | | | | | | | | | | |
|---|---|---|---|---|---|---|---|---|---|---|---|---|---|
|  |  | $V$ | | | $\delta$ | | | $\varepsilon$ | | | $v^\perp$ | | |
|  |  | BG | Ai | At | BG | Ai | At | BG | Ai | At | BG | Ai | At |
| Step 1 | 30 – 0.7 | 0.7 | 26.7 | 2.7 | - | - | - | 5.4 | 33.0 | 45.7 | 0.5 | 21.0 | 6.4 |
| Step 2 | 0.7 – 0.5 | 0.7 | 26.2 | 2.5 | 1.8 | 17.4 | 14.9 | 5.7 | 29.8 | 41.8 | 0.5 | 21.5 | 6.1 |

**Table 6: Same as Table 5 but for the two-step sequential inversion of P[$v^\perp$] (Fig. 7). Model fits are similar to those achieved by the simultaneous inversion of this parametrization: $v$ and $v^\perp$ are well retrieved, while recovery of $\delta$ and $\varepsilon$ is unsuccessful. Final data fit is identical to the one obtained by simultaneously inverting for P[$v^\perp$].**





| | Residuals RMS (ms) | Mean relative misfits (%) | | | | | | | | | | | | |
|---|---|---|---|---|---|---|---|---|---|---|---|---|---|---|
| | | $v$ | | | $\delta$ | | | $\varepsilon$ | | | $v^{\perp}$ | | |
| | | BG | Ai | At | BG | Ai | At | BG | Ai | At | BG | Ai | At |
| P[$\varepsilon$] | 19 – 0.1 | 0.1 | 25.2 | 0.3 | 0.8 | 15.4 | 7.7 | 0.3 | 24.4 | 0.5 | 0.1 | 29.4 | 0.3 |
| P[$v^{\perp}$] | 21 – 0.4 | 0.2 | 24.2 | 1.1 | 1.4 | 30.6 | 9.9 | 1.2 | 31.9 | 7.4 | 0.1 | 29.7 | 0.5 |

**Table 7: Same as Table 4 but for the simultaneous inversion of P[$\varepsilon$] and P[$v^{\perp}$] using $v$ and $\varepsilon$ or $v^{\perp}$ target models as initial models (Figs 8 and 9). Model fits for $v$ and $\varepsilon$ or $v^{\perp}$ are close to perfect as expected, but even so the recovery of $\delta$ is partial at most. Data**
**misfit is smaller than for the original inversions in Figs 4 and 5. P[$\varepsilon$] yields a better result according to all indicators, and as for all previous tests, the recovery of $v^{\perp}$ from P[$\varepsilon$] is notably better than that of $\varepsilon$ from P[$v^{\perp}$].**




| | Residuals RMS (ms) | Mean relative misfits (%) | | | | | | | | | | | |
|---|---|---|---|---|---|---|---|---|---|---|---|---|---|
| | | $v$ | | | $\delta$ | | | $\varepsilon$ | | | $v^\perp$ | | |
| | | BG | Ai | At | BG | Ai | At | BG | Ai | At | BG | Ai | At |
| P[$\varepsilon$] | 61 – 0.8 | 3.3 | 22.0 | 3.1 | - | - | - | 14.6 | 28.6 | 5.2 | 2.6 | 26.8 | 3.2 |
| P[$v^\perp$] | 61 – 0.8 | 3.5 | 21.1 | 3.3 | - | - | - | 24.7 | 40.7 | 15.2 | 2.0 | 27.8 | 2.4 |

**Table 8: Same as Table 4 but for the simultaneous inversion of P[$\varepsilon$] and P[$v^\perp$] following Eq. (13), i.e. neglecting $\delta$ in Eqs (1) and (3). Model fits for $v$, $\varepsilon$, and $v^\perp$, as well as data misfit are all significantly worse than for any of the previous tests.**





| $\delta$ value for initial model | $P[\varepsilon]$ | | $P[v^\perp]$ | |
|---|---|---|---|---|
| | Initial and final traveltime residuals RMS (ms) | Mean ± mean deviation of inverted $\delta$ model | Initial and final traveltime residuals RMS (ms) | Mean ± mean deviation of inverted $\delta$ model |
| 0.1 | 39 – 11 | 0.13 ± 0.05 | 39 – 5 | 0.13 ± 0.04 |
| 0.11 | 37 – 8 | 0.13 ± 0.04 | 37 – 4 | 0.14 ± 0.03 |
| 0.12 | 35 – 4 | 0.15 ± 0.03 | 35 – 0.5 | 0.14 ± 0.02 |
| 0.13 | 34 – 0.5 | 0.15 ± 0.01 | 34 – 0.5 | 0.14 ± 0.02 |
| 0.14 | 32 – 0.4 | 0.15 ± 0.01 | 32 – 0.5 | 0.15 ± 0.01 |
| 0.15 | 31 – 0.4 | 0.16 ± 0.01 | 31 – 0.5 | 0.16 ± 0.01 |
| 0.16 | 30 – 0.4 | 0.160 ± 0.008 | 30 – 0.5 | 0.160 ± 0.008 |
| 0.17 | 28 – 0.4 | 0.16 ± 0.01 | 29 – 0.6 | 0.17 ± 0.01 |
| 0.18 | 27 – 0.4 | 0.17 ± 0.01 | 28 – 0.5 | 0.17 ± 0.01 |
| 0.19 | 27 – 0.5 | 0.17 ± 0.01 | 27 – 0.5 | 0.18 ± 0.01 |
| 0.2 | 27 – 0.5 | 0.18 ± 0.02 | 27 – 0.5 | 0.18 ± 0.02 |
| 0.21 | 27 – 0.5 | 0.18 ± 0.02 | 27 – 0.5 | 0.19 ± 0.02 |
| 0.22 | 27 – 0.5 | 0.19 ± 0.02 | 27 – 0.5 | 0.19 ± 0.02 |
| 0.23 | 27 – 5 | 0.20 ± 0.04 | 27 – 2 | 0.20 ± 0.03 |
| 0.24 | 28 – 11 | 0.22 ± 0.06 | 28 – 5 | 0.20 ± 0.04 |

**Table 9: Results of the procedure to approximate an initial $\delta$ model. The RMS of the final traveltime residuals shows a clear change in order of magnitude in the subranges (0.13,0.22) and (0.12,0.22) depending on the parameterization. Results for initial $\delta$ = 0.16 correspond to the examples in Figs 4 and 5.**





**Figures**

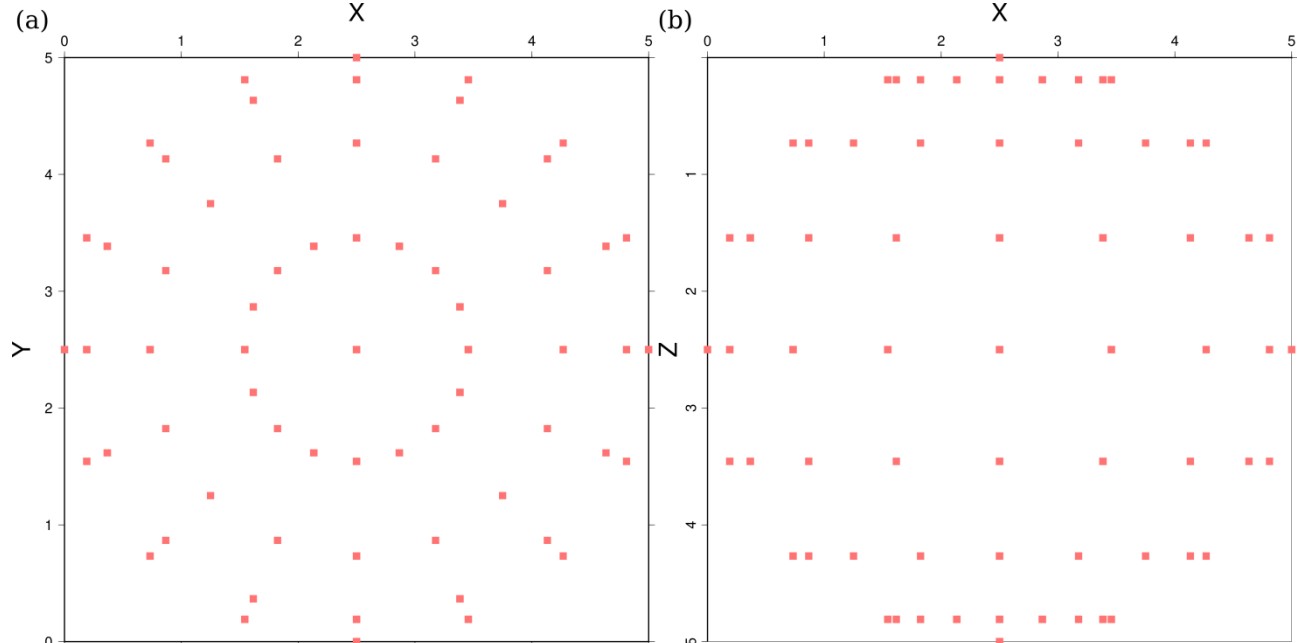

**Figure 1: (a) Horizontal and (b) vertical views of the acquisition geometry for the inversion tests. 114 sources and 114 receivers (red boxes) are located at 2.5 km from the center of the model, at the locus defined by the surface of the sphere inscribed in the cube, and placed at the crossing points of 16 meridians with 7 parallels, and at each pole. Thus, in the inversion tests we used**
**12,882 traveltimes from 114 sources each recorded at 113 receivers, i.e. all receivers record all sources, except for the one coinciding in location. The acquisition geometry for the accuracy and sensitivity tests is similar, only in this case for sources and receivers at the crossing points of 32 meridians and 15 parallels, plus one of each at the two poles, and using just 482 traveltimes from diametrically-opposed source–receiver pairs arranged, i.e. each receiver exclusively records the first arrival traveltime from its paired source.**




**Figure 2: Relative traveltime misfits in percentage with respect to the analytic value for each of the four possible anomalies separately, and for each of the two selected meridians, 0 rad (left) and π/4 rad (right) azimuths. Polar angle origin is in the downward vertical axis. Mean values and deviations are shown in Table 2.**





**Figure 3: Sensitivities for the meridians at azimuths 0 rad (left) and π/4 rad (right) as a function of the polar angle (origin in the downward vertical axis). (a) and (b) Synthetic relative sensitivities in percentage, (c) and (d) synthetic normalized sensitivities, and (e) normalized analytic sensitivities. Sensitivity values displayed correspond to all source – receiver pairs along the selected meridians of the acquisition configuration.**




**Figure 4: Simultaneous inversion with P[ε]. Horizontal slices of the relative differences between target and initial (first row), final and initial (second row), and target and final (third row) models at 2.5-km depth for the four parameters. $v^\perp$ is derived from Eq. (2). The range of the color scale for $v^\perp$ is wider than for the rest of parameters because the heterogeneity is calculated considering the 25% anomalies in $v$ and $ε$, which yields a ~29.3% anomaly in $v^\perp$. First and second row would be identical if the inversion were**

**perfect, whereas the third row would display a homogeneous value of 0%. The quality of the recovery of each parameter is in correlation with their sensitivities (Fig. 3). Recovery of $v$ is satisfactory with anomaly values close to the target and well-defined anomaly boundaries. $ε$ recovery is partial, the anomaly is centered but its magnitude and shape are not as accurate as in the case of both velocities; even so it allows for a successful recovery of $v^\perp$ through Eq. (2), both in anomaly magnitude and shape. As for $δ$, recovery is unsuccessful.**




**Figure 5:** Same as Fig. 4 but with P[$\nu^\perp$]. $\varepsilon$ is derived from Eq. (2). The quality of the recovery is in correlation with sensitivity (Fig. 3). Both velocities are satisfactorily recovered. The magnitude of the anomaly in $\nu$ is better recovered than for P[$\varepsilon$], whereas the opposite occurs for $\nu^\perp$. Anomaly boundaries for both velocities are not as well determined as for P[$\varepsilon$]. $\varepsilon$ and $\delta$ are not recovered.




**Figure 6: Same as Fig. 4 but for the two-step sequential inversion strategy with P[$\varepsilon$]. In the first step only $\delta$ was fixed. In the second step, only $\varepsilon$ was fixed. Final models from step 1 were used as initial model for step 2. $v$ (1st column) is well recovered in step 1 (top panel), and it is barely modified by the second step (middle and bottom panels). $\delta$ is fixed to the initial homogeneous model in step 1, and its recovery is unsuccessful in step 2. Recovery of $\varepsilon$ is limited compared to $v$, but significantly better than that of $\delta$. Nonetheless, it proves to be good enough to provide a satisfactory recovery of $v^{\perp}$ using Eq. (2).**





**Figure 7: Same as Fig. 6 but for the two-step sequential inversion strategy with P[$v^{\perp}$]. In the first step only $\delta$ is fixed, whereas in the second one all parameters are inverted. $v$ (1st column) and $v^{\perp}$ (4th column) are well recovered in step 1 (top panel), and they are barely modified by the second step (middle and bottom panels). $\delta$ and $\varepsilon$ are not properly recovered; in both cases some sort of irregular perturbations approximately centered in the cube are retrieved but bearing no resemblance to the target anomalies.**







**Figure 8: Same as Fig. 4 but using target models as initial models for all parameters in P[ε] but δ. This test was conducted to study the recovery of δ under unrealistically optimal circumstances, and even with these perfect initial conditions, recovery is, at best, extremely complicated due to the small sensitivity (Fig. 3); magnitude and shape are only partially recovered. In the first row, differences for ν and ε are 0% since we use target models as initial ones. For this same reason, the second and third rows show that differences between target and inverted for these three parameters are hardly observable, indicating that inversion is not modifying ν and ε even though they are not fixed. Consequently, the resulting recovery of ν⊥ through Eq. (2) is almost perfect as well.**



**Figure 9: Same as Fig. 8 but using target models as initial models for all parameters in P[$v^\perp$] except for $\delta$. The magnitude of the $\delta$ anomaly is better recovered than for P[$\varepsilon$], but the shape is not as well retrieved, and artifacts appear in the background area. Differences between inverted and target $v$ and $v^\perp$ models are still hardly observable but not as much as for $v$ and $\varepsilon$ in the case of P[$\varepsilon$], and thus the recovery of $\varepsilon$ through Eq. (2) is also not as good.**