# Peer review of "Anisotropic P-wave traveltime tomography implementing Thomsen's weak approximation in TOMO3D"

_Solid Earth, 2019_

## Referee Comment (RC1) · Anonymous Referee #1 · 1 Apr 2019

__Overall appreciation__

This paper presents an extension to the code TOMO3D to allow inverting travel times for transverse anisotropic media. The accuracy of the forward operator is evaluated by comparing the output with an analytic solution, and the sensitivity is also quantified. The main part of the paper contains the results of a series of inversion tests to assess the performance of the proposed approaches. Overall the paper is well structured and follows a logical reasoning. There are however a number of points that should be addressed, as detailed in the following section.

__Specific points__

1 - Does the paper address relevant scientific questions within the scope of SE?

Yes, very few codes exist for travel time inversion in anisotropic media and I found interesting the idea of evaluating the sensitivity as function of anisotropy parameters. I also welcome the fact that the models retrieved from inversion are compared quantitatively with the true model (too often a qualitative appreciation is presented).

2- Does the paper present novel concepts, ideas, tools, or data?

The paper is pretty classic in its form and approach. An extension to an existing code is presented, that allows inverting travel time data for anisotropy parameters.

3- Are substantial conclusions reached?

Partly. Tests were done with synthetic data, which allow evaluating quantitatively the performance of the inversion. However, I think that the conclusions are not fully supported by the presented work, this for three reasons. First, the authors did not study the influence of noise on the robustness of the results. Inevitably, noise is present in field data (picking accuracy, timing accuracy, statics, etc) and at least the effect of some gaussian noise should be investigated with the synthetic data.

Second, the data acquisition geometry could never be achieved in reality. For the presented tests, an anomalous spheric body is surrounded by sources (Tx) and receivers (Rx) in the whole space. This geometry supposes first that the location of the anomalous body is known, and second that access is possible underground almost everywhere around the body. At best, surface and a few borehole Tx & Rx are typically available for typical surveys. For such geometries, the forward operator does not allow uniform resolution, such as illustrated in the results of the paper. So I think that the capacity to resolve the anomaly is over optimistic.

Third, the initial model is quite close to the true model (the initial model is equal to the background of the true model). I don't think that it is realistic to know that well the properties of the background, especially when the background itself is anisotropic.

How could one know the anisotropy parameters epsilon & delta of the background?

4- Are the scientific methods and assumptions valid and clearly outlined?

Some assumptions are not realistic (see point above).

5- Are the results sufficient to support the interpretations and conclusions?

See point 3.

6- Is the description of experiments and calculations sufficiently complete and precise to allow their reproduction by fellow scientists (traceability of results)?

Yes, I see not problem for this point.

7- Do the authors give proper credit to related work and clearly indicate their own new/original contribution?

Yes

8- Does the title clearly reflect the contents of the paper?

Yes

9- Does the abstract provide a concise and complete summary?

Yes

10- Is the overall presentation well structured and clear?

Overall yes, but the section describing the inversion strategies is somewhat hard to follow (especially the details of the sequential inversion). I think that a figure with flowcharts could help.

11- Is the language fluent and precise?

Yes

12- Are mathematical formulae, symbols, abbreviations, and units correctly defined and

used?

Overall yes, but I think that the anisotropy parameters epsilon & delta should be formally defined in section 2. Some parameters in equation 4 are not defined.

13- Should any parts of the paper (text, formulae, figures, tables) be clarified, reduced, combined, or eliminated?

Perhaps the terms for the reflected rays could be removed from the equations (the case of reflected rays is not treated in the paper).

14- Are the number and quality of references appropriate?

Yes, but some references cites in the text are not in the list at end of the paper.

Other comments can be found in the annotated version of the paper.

Please also note the supplement to this comment:
https://www.solid-earth-discuss.net/se-2019-44/se-2019-44-RC1-supplement.pdf

———————————————

---

## Referee Comment (RC2) · Anonymous Referee #2 · 22 Apr 2019

**General comments**

The authors present a modification to TOMO3D seismic tomographic inversion code which allows to include anisotropy parameters (assuming VTI symmetry) in the inversion. Such code is a valuable and interesting contribution, as most of the existing tomographic codes assume an isotropic medium. The manuscript presents necessary evaluation of reliability of the code. Formally, the structure of the paper is correct, it contains all necessary parts and is generally well written, although some parts (details below) are hard to follow. The main goal of the manuscript is to evaluate three characteristics of the code/method: the accuracy of the forward code, sensitivity of the method

for anomalies of model parameters and resolving ability of the inversion. In my opinion, the papers gives satisfactory answer for first two questions are basically satisfactory, while the third question is studied only partially (details below). An interesting outcome of the inversion tests is the discussion about poor sensitivity of the traveltime data on the $\delta$ parameter, even in case of 'ideal' measurements geometry (spherical geometry and uniform angular spacing of sources/receivers locations).

**Specific comments**

From the parametrization used (Thomsen parameters) it implicitly follows that the VTI symmetry of the medium is assumed (also, the inversion does not solve for orientation of the symmetry axis, so I guess it is assumed to be vertical) but it is not clearly stated in the text, and statements like 'anisotropic tomography' (also in the title!) suggest at the first glance that the code can be used for media with more general and more complicated symmetries. If I'm right about this, I would recommend to state it explicitly that the code can model VTI media only. This would make clear that it cannot be used e.g. for modeling of azimuthal anisotropy (which is usually due to HTI or TTI medium).

L120: The feasibility of determination of the $\delta$ parameter is widely discussed in the text, but the definition of the parameter itself is not given. I think it should be added to the text.

L128-132: I understand the relationship between anisotropy parameters and Vnmo, but the comment and conclusion about the Vnmo in L128-132 is unclear for me.

-Accuracy and sensitivity tests:

Fig. 3: what is the meaning of the 'normalized difference'? And why 'relative difference' for v is constant for all angles, while 'normalized difference' is sinusoidal?

L198: The authors wrote "..for each of the four parameters. . .25% anomaly was added, while . . . the rest of parameters was homogeneous" – This seems to be contradictory (or it is explained in a misleading way). Parameters v, vT (I use vT instead of the

symbol used in the manuscript for v in perpendicular direction) and $\varepsilon$ are interrelated by Eq. 2, so changing vT and keeping v constant must change also $\varepsilon$ – in result, two parameters are changed. I know that this is not a problem for inversion, as parameters vT and $\varepsilon$ are used alternatively, not in the same time, but such description introduces confusion in interpretation of the sensitivity test results. Also, even if anomalies of every parameter are the same (25%), we can hardly compare cases of $\varepsilon$ and vT anomalies: - 25% $\varepsilon$ anomaly results in value of 0.2 (compared to background 0.16), while: - for vT, the same anomaly (25%) with respect to background 2.32 results in vT=2.9. This, connected with v in the anomaly being 2.0 (undisturbed, so equal to background), results in $\varepsilon$=0.45 –an anomaly few times larger than in previous case. (This is probably the reason why in Fig. 3 sensitivity for $\varepsilon$ reaches ∼0.7%, while sensitivity of vT is few times larger and reaches 4.5%)

-Inversion tests:

The inversion tests are done properly, assuming several variants and various strategies, and results (especially concerning the poor recovery of $\delta$ parameter) are interesting, but their main drawback is assumption of near-ideal experimental conditions: - the measurements geometry (spherical geometry and uniform angular spacing of sources/receivers locations, which results in unrealistically uniform ray coverage). - no noise assumed. Such conditions are almost never possible in case of seismic in-situ experiments, where sources and receivers locations are usually limited to the earth surface, resulting in quite unfavorable ray geometry for solving inversion problem. Therefore, presented tests provide good estimate of 'maximum capabilities' of the code. For the case of modeling the VTI medium in ideal conditions (which is valuable because it shows 'weak points' of the code and parametrization assumed – if the method fails in some aspect in ideal conditions, it will fail even more in case of real data). But such tests give no or very little information about reliability (expected resolution, dependence on the noise level, dependence on the initial model etc.) of a typical seismic experiment.

I think that the manuscript would improve a lot if the authors could add more realistic synthetic tests: in order to properly check behavior of the code in case of typical data form seismic experiment, the tests should assume noisy data, surface location of sources/receivers, and also the dependence of the result on various initial models should be studied.

Figs 4-9: It should be marked which column represents which parameter.

**Summarizing, I think that a moderate/major revision, taking into account the above comments, is needed.**

---

## Author Comment (AC1) · 28 Jun 2019

Please see the pdf document attached as a supplement in the link below. It contains, in this same order, the new manuscript with tracking of changes, the supplementary material, and the responses to both RC1 and RC2.

Please also note the supplement to this comment:
https://www.solid-earth-discuss.net/se-2019-44/se-2019-44-AC1-supplement.pdf

---

## Author Comment (AC2) · 28 Jun 2019

[revised manuscript text omitted]

**Table 7: Same as Table 4 but for the simultaneous inversion of P[ε] and P[$v^\perp$] using $v$ and ε or $v^\perp$ target models as initial models (Figs 9 and 10). Model fits for $v$ and ε or $v^\perp$ are close to perfect as expected, but even so the recovery of δ is partial at most. Data misfit is smaller than for the original inversions in Figs 5 and 6. P[ε] yields a better result according to all indicators, and as for all previous tests, the recovery of $v^\perp$ from P[ε] is notably better than that of ε from P[$v^\perp$].**

| | Residuals RMS (ms) | Mean relative  differences (%) | | | | | | | | | | | |
|---|---|---|---|---|---|---|---|---|---|---|---|---|---|
| | | *v* | | | δ | | | ε | | | $v^\perp$ | | |
| | | BG | AI~i~ | AT~t~ | BG | AI~i~ | AT~t~ | BG | AI~i~ | AT~t~ | BG | AI~i~ | AT~t~ |
| P[ε] | 61 – 0.8 | 3.3 | 22.0 | 3.1 | - | - | - | 14.6 | 28.6 | 5.2 | 2.6 | 26.8 | 3.2 |
| P[$v^\perp$] | 61 – 0.8 | 3.5 | 21.1 | 3.3 | - | - | - | 24.7 | 40.7 | 15.2 | 2.0 | 27.8 | 2.4 |

**Table 8: Same as Table 4 but for the simultaneous inversion of P[ε] and P[$v^\perp$] following Eq. (11), i.e. neglecting δ in Eqs (1) and (3). Model  differences for $v$, ε, and $v^\perp$, as well as data misfit are all significantly worse than for any of the previous tests.**

[revised manuscript text omitted]

765 **Figure S2: For the π/4 rad meridian, relative traveltime errors in percentage with respect to the analytic value for three pairs of equivalent simulations used in the sensitivity tests. Polar angle origin is in the downward vertical axis. Both parametrizations produce almost identical accuracies.**

*Mathematical proof for the shapes of v sensitivity in both parametrizations*

770 Regarding the shapes of relative and normalized sensitivities for v, in the following we provide mathematical proof showing

that the former is constant whereas the latter is sinusoidal.

We use $\Delta t$ to refer to the difference between the travel times measured with and without the 25% anomaly. $x_A$ is the thickness

of the anomaly whereas $x_B$ is the total ray path length from source to receiver. $v_A$ and $v_B$ are the anisotropic velocities in the

anomaly and in the background respectively.

$$\Delta t = \Delta\left(\frac{x}{v}\right) = \frac{x_B}{v_B} - \left(\frac{x_A}{v_A} + \frac{x_B - x_A}{v_B}\right) = \frac{x_A(v_A - v_B)}{v_A v_B}$$

775 According to the definition we have just given, the sensitivity expressed as relative difference is

$$S_R = \frac{\frac{x_A(v_A - v_B)}{v_A v_B}}{t_R}$$

where $t_R$ is the travel time measured without the anomaly

$$t_R = x_B / v_B$$

With that $S_R$ becomes

$$S_R = \frac{x_A}{x_B}\left(1 - \frac{v_B}{v_A}\right)$$

In the case of a 25% anomaly in the $v$ parameter in P[$\varepsilon$], $v_A$ and $v_B$ are as follows

$$v_A = v_{PA}(1 + \delta\,\sin^2\theta\,\cos^2\theta + \varepsilon\,\sin^4\theta)$$
$$v_B = v_{PB}(1 + \delta\,\sin^2\theta\,\cos^2\theta + \varepsilon\,\sin^4\theta)$$

We use $v_{PA}$ and $v_{PB}$ to refer to the axis-parallel velocity parameter $v$. For a 25% anomaly in $v$ their proportion is

$$v_{PA} = 1.25\,v_{PB}$$

780 The values for $x_A$ and $x_B$ are

$$x_A = 1\ km \text{ and } x_B = \ km$$

Thus, the final expression for $S_R$ for a 25% anomaly in $v$ in P[$\varepsilon$] is a constant 0.05 (or 5%).

$$S_R = \frac{1}{5}\left(1 - \frac{1}{1.25}\right) = 0.05$$

Regarding the sensitivity expressed as normalized difference, the general expression for an anomaly in any of the four

parameters is

$$S_N = \frac{\frac{x_A(v_A - v_B)}{v_A v_B}}{\Delta t_{MAX}} = \frac{x_A}{\Delta t_{MAX}}\left(\frac{1}{v_B} - \frac{1}{v_A}\right)$$

where $\Delta t_{\text{MAX}}$ is the maximum travel time difference among all four parameters. This expression will contain some combination of sinusoidal functions for all four possible anomalies. The sensitivity (normalized and relative) pattern for $v$ in $P[v^\perp]$ is different than for $P[\varepsilon]$ although it follows the same sinusoidal pattern and it has equal maxima. We can see that in the case of $P[v^\perp]$ both $S_R$ and $S_N$ will display a sinusoidal shape. Now $v_A$ and $v_B$ are

$$v_A = v_{PA}\left(1 + \delta\, sin^2\theta\, cos^2\theta + \left(\frac{v^\perp}{v_{PA}} - 1\right)sin^4\theta\right)$$

$$v_B = v_{PB}\left(1 + \delta\, sin^2\theta\, cos^2\theta + \left(\frac{v^\perp}{v_{PB}} - 1\right)sin^4\theta\right)$$

and the sine and cosine functions do not cancel out in $S_R$, as they did in the case of an anomaly in $v$ in $P[\varepsilon]$, nor in $S_N$. For the latter it is trivial to see that the sinusoidal dependencies are identical to the case in $P[\varepsilon]$.

Response to RC1

 *Comments in file se-2019-44-RC1.pdf*

3- Are substantial conclusions reached?

Partly. Tests were done with synthetic data, which allow evaluating quantitatively the performance of the inversion. However, I think that the conclusions are not fully supported by the presented work, this for three reasons. First, the authors did not study the influence of noise on the robustness of the results. Inevitably, noise is present in field data (picking accuracy, timing accuracy, statics, etc) and at least the effect of some gaussian noise should be investigated with the synthetic data. Second, the data acquisition geometry could never be achieved in reality. For the presented tests, an anomalous spheric body is surrounded by sources (Tx) and receivers (Rx) in the whole space. This geometry supposes first that the location of the anomalous body is known, and second that access is possible underground almost everywhere around the body. At best, surface and a few borehole Tx & Rx are typically available for typical surveys. For such geometries, the forward operator does not allow uniform resolution, such as illustrated in the results of the paper. So I think that the capacity to resolve the anomaly is over optimistic. Third, the initial model is quite close to the true model (the initial model is equal to the background of the true model). I don't think that it is realistic to know that well the properties of the background, especially when the background itself is anisotropic.

(*) We are aware that the synthetic experiment that we designed to perform our tests is not realistic. In fact, our goal is to use this canonical benchmark to assess accuracy, sensitivity, and performance of the four inversion strategies under ideal and equal conditions for all anisotropic parameters, avoiding the specificities and biases of a synthetic experiment simulating a particular field case study. These tests on a canonical benchmark provide conclusions that are generally informative of the code's performance. In other words, here we are not interested in the performance of the code in a more or less specific geological and experimental context, but in obtaining an upper limit to the code's capabilities, an ideal but generalizable estimation for the code's performance.

We have modified the manuscript to clarify the purpose of our testing approach and the reasons behind it. Specifically, we have edited the Abstract (lines 10 and 11), the third paragraph in the Introduction, the first paragraph to section 3 Synthetic tests, and the second paragraph in the Conclusions.

Nonetheless, in an upcoming paper presenting an application of the code to an anisotropic field data case, we will first evaluate the code's performance on a realistic synthetic experiment simulating this particular field study. This simulation should yield an estimation of the potential quality of the results as well as information on the best modeling strategy to approach this specific case, and it will include noise, an initial model obtained from isotropic tomography, and will replicate the same exact acquisition geometry used in the field. By synthetically reproducing a more or less specific type of seismic

experiment under some realistic circumstances, e.g. a certain noise level, we will get an idea of what to expect in that particular case with the selected noise level, receiver and source densities and distributions, geological features and anomalies, a priori information available in the initial models, etc. Therefore, this sort of realistic synthetic testing and the

830 conclusions that can be drawn from it become relevant and meaningful as preliminary work linked to a particular field data application.

We believe that adding these other type of tests here would result in an exceedingly long manuscript covering too many aspects. Also, we think that realistic synthetic tests are better presented along with the field data that they are simulating.

835 Thus, we prefer to separate our work into two publications, this first one of technical and methodological content, and a second one focusing on a field data application. Figure R1 corresponds to this anisotropic field case. In Sallarès et al. [2013] we obtained an isotropic Vp model from a refraction and wide-angle reflection seismic (WAS) data set (sub-horizontal propagation) and compared it to the image obtained from multichannel seismic (MCS) reflection data (near-vertical propagation). The top plot shows the isotropic Vp model, with the vertical coordinate converted from depth to two-way time

840 (TWT), superimposed onto the MCS image. The white circles inside the model delineate the geometry of inter-plate reflection imaged by MCS data. The thick red line corresponds to the TWT-converted inter-plate boundary obtained from WAS data. The mismatch between the two locations of the inter-plate boundary is most likely due to some degree of seismic anisotropy between near-vertical and sub-horizontal propagations. In the bottom plot we increase the Vp values by 15%, and with this the MCS and WAS locations of the inter-plate boundary now display a good match. Thus, this 15% increase is a

845 good initial estimate of $\varepsilon$, and it indicates, as a general trend, that near-vertical propagation is ~15% faster than sub-horizontal propagation in this area of the subsurface ($\varepsilon \approx 0.15$). For an estimate of $\delta$ we will use the $V_{NMO}$ model from the normal move-out correction of MCS data processing and the isotropic Vp model.

4- Are the scientific methods and assumptions valid and clearly outlined?

850 Some assumptions are not realistic (see point above).

See answer to point 3.

5- Are the results sufficient to support the interpretations and conclusions?

See point 3.

855 See answer to point 3.

10- Is the overall presentation well structured and clear?

Overall yes, but the section describing the inversion strategies is somewhat hard to follow (especially the details of the sequential inversion). I think that a figure with flowcharts could help.

860 We added a figure depicting the flowchart for the two-step sequential inversion (new Figure 4).

12- Are mathematical formulae, symbols, abbreviations, and units correctly defined and used?

Overall yes, but I think that the anisotropy parameters epsilon & delta should be formally defined in section 2. Some parameters in equation 4 are not defined.

865 (#) The definitions of $\delta$ and $\varepsilon$ are available in Thomsen [1986] which is the main reference for our work and for weak VTI anisotropy in general. We chose to focus on our formulation, and we only reproduced the formula for anistropic Vp because it is at the core of our anisotropic modeling tool. However if the Editor deems it necessary to include these definitions in our manuscript we will do so. Also we corrected the definitions for Equation 4.

870 13- Should any parts of the paper (text, formulae, figures, tables) be clarified, reduced, combined, or eliminated?

Perhaps the terms for the reflected rays could be removed from the equations (the case of reflected rays is not treated in the paper).

We removed Equations 11 and 12 as well as the terms related to reflections and interface depth in Equation 3. They are not used in this paper, and we will discuss them in the upcoming article presenting an application to anisotropic field data.

875

14- Are the number and quality of references appropriate?

Yes, but some references cites in the text are not in the list at end of the paper.

We have added the missing citations to the reference list. Lines 596 to 598, 605, and 606.

880 *Comments in se-2019-44-RC1-supplement.pdf*

Section 2 Modelling anisotropy

This section should contain the mathematical definition of epsilon and delta.

See answer marked with (#).

885

Subsection 2.2 Anisotropy in TOMO3D: Thomsen's weak anisotropy approximation

Three cited publications missing in the reference list.

We added the three missing citations to the reference section. See answer to point 14.

890 Do you assign the properties within the cells or at the grid nodes? e.g. is the number of unknown parameters a function of the number of cells or the number of nodes?

Properties are assigned to the grid nodes, so that the number of unknowns is a function of the number of nodes. We have modified the text to clarify this point (lines 108 and 109).

895 Equation 3

Delta z not defined.

We removed the parameters related to reflected rays and reflector depth from Equation 3, which includes $\Delta z$, since they are not used in this paper. They will be discussed in the field data application paper.

900 Equation 4

r_m not defined.

The definition of $r_m$ was already in the text (line 165) although the subscript m was missing. However, we have corrected lines 162 to 167 to clarify the description of Equation 4.

905 Section 3 Synthetic tests, line 195

You mean the values of the v, epsilon and delta parameters at grid nodes?

Yes. We have edited this phrase to clarify its meaning (line 215).

Subsection 3.1 Accuracy, lines 198 and 199

910 The body will not eventually contain anomalous v, epsilon and delta?

Yes, and that is what we consider when performing the inversion tests. However, regarding accuracy and sensitivity we wish to evaluate each parameter separately so that we can compare the code's performances for each of them. We have edited and reordered sections 3.1 and 3.2 to stress this point, and also as a result of a comment by the other referee.

915 Lines 203 to 205

This* is totally unrealistic, how do you expect that your conclusions will be valid for real life scenarios? [*referring to the acquisition geometry]

As mentioned in the answer marked with (*), our goal is to obtain an upper limit to the code's performance, in this case regarding the accuracy for each parameter. The conclusions are valid in the sense that they inform us on the limits of our

920 modeling tool. We do not pretend in any way that the canonical benchmark model or acquisition used simulate any particular field study.

Line 209

When computed with respect to the analytic solution, this is a relative error. A misfit is generally the difference with an

925 observation.

We replaced "misfits" by "errors" here and elsewhere in the manuscript when referring to the relative difference between calculated and analytic traveltimes.

Subsection 3.2 Sensitivity, line 232

930   Would be interesting to see if the relationship is linear with respect to changes in epsilon and delta.

Figure R2 shows the analytic $\delta$ sensitivities for anomalies of 10%, 15%, 20%, and 25%. The linear relationship between sensitivity and anomaly increment is particularly clear observing the evolution of the maxima. Given the form of Equation 1, one can infer that $\varepsilon$ sensitivity will have an analogous behavior.

935   Subsection 3.3 Inversion results, line 250

How can one expect to know the anisotropy parameters of the background? Moreover, you did not study the effect of noise on the robustness of the inversion.

(^) There exist compilations of values for Thomsen's anisotropy parameters (e.g. Thomsen [1986] for sedimentary rocks and related materials) or for the components of the elastic modulus tensor (e.g. Almqvist and Mainprice [2017] for continental
940   crust rocks) from which Thomsen's parameters can be derived. Also, we can extract estimates of $\delta$ from multichannel reflection data processing, specifically using the mathematical relationship between $v$, $\delta$, and $V_{NMO}$, and an isotropic estimate of $v$. An estimate for $\varepsilon$ can be obtained by comparing velocity models derived from multichannel and wide-angle seismic data since they are representative of near-vertical and sub-horizontal propagations respectively, and $\varepsilon$ represents the relative difference between them (Fig. R1). We will detail and conduct these procedures to estimate $\delta$ and $\varepsilon$ for initial model building
945   in field data applications in the upcoming paper.

As discussed in answer (*), we assume that we know this background value perfectly in order to evaluate the code's performance in retrieving an anomalous body within this background in what is an ideal situation providing an upper-limit but generalizable estimation of the code's capabilities and indicating potential weaknesses. For identical reasons we do not
950   include noise in our data.

Lines 254 and 255

Not clear [referring to the description of the acquisition geometry].

We edited lines 321 to 326 to clarify the acquisition geometry.

955

Lines 277 to 279

This* is quite confusing, use a flowchart for each [*referring to the description of the sequential inversion].

We added a figure with a flowchart reproducing the two-step sequential inversion strategy (new Figure 4).

960   Section 4 Discussion, line 368

The problem is actually to obtain this a priori information.

As mentioned in answer (^), it is possible to obtain reasonable estimates of the anisotropy parameters.

Lines 376 to 378

965     What is the basis for this statement? [referring to "it would probably be recommendable to use a coarser discretization for $\delta$ than for the other parameters, and in general a finer discretization for the more sensitive parameters (Fig. S1)"].

The heterogeneities that can be resolved for a particular parameter depend on its sensitivity. The more sensitive a parameter is, the smaller the spatial scale and the relative variation of the resolvable heterogeneities. An heterogeneity of a given scale and variation will produce a greater effect on data for a parameter of greater sensitivity. Thus, for a parameter of greater

970 sensitivity, it will be easier for the code to identify smaller heterogeneities both in scale and variation, which will require a finer grid. We have modified the manuscript to clarify this point (lines 461 to 464).

Section Author contribution

Data acquisition? There is no data in that paper.

975 "Funding acquisition" means "acquisition of the financial support for the project leading to this publication" as defined in section The CRediT Roles at https://www.casrai.org/credit.html. We have replaced "funding acquisition" by "acquisition of financial support".

Table 4

980 The opposite [referring to the ideal values for AI and AT in "AI and AT indicate the resemblance between true and recovered anomalies, and their ideal values are 0% and 25% (~29.3% in the case of v ⊥ ) respectively"].

Corrected.

Table 5 (to Table 8)

985 Keep consistent notation [referring to the systematic use of either AI or Ai, and AT or At].

Corrected.

Response to RC2

 *Comments in file se-2019-44-RC2.pdf*

General comments

The authors present a modification to TOMO3D seismic tomographic inversion code which allows to include anisotropy parameters (assuming VTI symmetry) in the inversion. Such code is a valuable and interesting contribution, as most of the 995 existing tomographic codes assume an isotropic medium. The manuscript presents necessary evaluation of reliability of the code. Formally, the structure of the paper is correct, it contains all necessary parts and is generally well written, although some parts (details below) are hard to follow. The main goal of the manuscript is to evaluate three characteristics of the code/method: the accuracy of the forward code, sensitivity of the method for anomalies of model parameters and resolving ability of the inversion. In my opinion, the papers gives satisfactory answer for first two questions are basically satisfactory, 1000 while the third question is studied only partially (details below). An interesting outcome of the inversion tests is the discussion about poor sensitivity of the travel time data on the $\delta$ parameter, even in case of 'ideal' measurements geometry (spherical geometry and uniform angular spacing of sources/receivers locations).

Specific comments

1005 From the parametrization used (Thomsen parameters) it implicitly follows that the VTI symmetry of the medium is assumed (also, the inversion does not solve for orientation of the symmetry axis, so I guess it is assumed to be vertical) but it is not clearly stated in the text, and statements like 'anisotropic tomography' (also in the title!) suggest at the first glance that the code can be used for media with more general and more complicated symmetries. If I'm right about this, I would recommend to state it explicitly that the code can model VTI media only. This would make clear that it cannot be used e.g. for modeling 1010 of azimuthal anisotropy (which is usually due to HTI or TTI medium).

The first line of the Abstract states that with this article "We present the implementation of Thomsen's weak anisotropy approximation for VTI media within TOMO3D". In the third paragraph of the Introduction we describe the two main objectives of this work pointing out that the we present "the anisotropic version of TOMO3D for the study of VTI weakly-anisotropic media". We believe that with these two statements it is clear that the anisotropic version of TOMO3D assumes 1015 VTI symmetry for the media, but we also added "VTI" to the title of subsection 2.2 that now reads "Anisotropy in TOMO3D: Thomsen's weak VTI anisotropy formulation", in line 69 in the introduction for section 2 Modelling anisotropy, and in the first paragraph of the Conclusions.

Regarding the title, we consider that mentioning Thomsen's weak approximation is sufficient since it implies that VTI 1020 symmetry is used, as the referee corroborates. However, we would not oppose changing it to "Anisotropic P-wave traveltime tomography implementing Thomsen's weak VTI approximation in TOMO3D".

L120: The feasibility of determination of the δ parameter is widely discussed in the text, but the definition of the parameter itself is not given. I think it should be added to the text.

The definition of $\delta$ is available in Thomsen [1986] which is the main reference for our work. We chose to focus on our formulation, and we only reproduced the formula for anistropic Vp because it is key to our anisotropic modeling tool. However if the Editor deems it necessary to include this definition in our manuscript we will do so.

L128-132: I understand the relationship between anisotropy parameters and Vnmo, but the comment and conclusion about the Vnmo in L128-132 is unclear for me.

However, a mathematical relationship between δ, v, and the normal move-out velocity (V NMO ) exists. V NMO models are built as part of the normal move-out correction in seismic reflection data processing. At best, our travel time tomographic method would be able to produce approximations of the actual V NMO models. Furthermore, such approximations would only be meaningful, if ever, when derived from travel times of a seismic reflection data set, for which the normal move-out correction and thus the V NMO are defined. Of course, in such a case, actual V NMO models would be obtained from the normal move-out correction, and therefore δ could be calculated provided that a v model is available, for instance from our travel time tomography. Thus, we only consider Eq. (2), and we implemented two parametrizations of the medium: (v, δ, ε) and (v, δ, v⊥).

The conclusion is that we do not consider $V_{NMO}$ to be a useful parameter in describing the anisotropic VTI media so that we do not implement parametrizations ($v$, $V_{NMO}$, $\varepsilon$) and ($v$, $V_{NMO}$, $v^{\perp}$). Unlike axis-parallel and axis-perpendicular propagation velocities, which define the physics of the medium, $V_{NMO}$ is a by-product of multichannel seismic reflection data processing, a mathematical construct that involves the assumptions of a stratified media with constant velocity layers and of small spread, i.e. near-vertical propagation. The best possible way to obtain a $V_{NMO}$ model is obviously as this by-product of the processing of the sort of data that it is defined for. It does not seem clever to try and obtain it by other means, less so if the data and modeling used to do so do not fulfill the necessary assumptions of the processing for which $V_{NMO}$ is defined. $\delta$ is needed to describe the weak VTI anisotropy as proposed by Thomsen [1986], whereas we do not need $V_{NMO}$.

We admit that the original text did not convey these ideas in a clear way, and so we have modified the text to clarify our points (paragraph starting at line 131).

-Accuracy and sensitivity tests:

Fig. 3: what is the meaning of the 'normalized difference'? And why 'relative difference' for v is constant for all angles, while 'normalized difference' is sinusoidal?

(^) Normalized difference is the difference between the travel times measured with and without anomaly divided by the greatest of these differences among all four parameters. Relative difference is the difference between the travel times

measured with and without anomaly divided by the travel time without anomaly (and multiplied by 100). These are the relative changes produced in travel times by an anomaly of 25% relative change in each particular parameter and as a function of the polar angle. We realize that these two manners of presenting sensitivity are not properly described in the text. We have modified it to include their definitions (first paragraph of new section 3.1 Sensitivity).

1060

Regarding the shapes of relative and normalized sensitivities for v, in the following we provide mathematical proof showing that the former is constant whereas the latter is sinusoidal (also in the supplementary material).

We use $\Delta t$ to refer to the difference between the travel times measured with and without the 25% anomaly. $x_A$ is the thickness
1065 of the anomaly whereas $x_B$ is the total ray path length from source to receiver. $v_A$ and $v_B$ are the anisotropic velocities in the anomaly and in the background respectively.

$$\Delta t = \Delta\left(\frac{x}{v}\right) = \frac{x_B}{v_B} - \left(\frac{x_A}{v_A} + \frac{x_B - x_A}{v_B}\right) = \frac{x_A(v_A - v_B)}{v_A v_B}$$

According to the definition we have just given, the sensitivity expressed as relative difference is

$$S_R = \frac{\dfrac{x_A(v_A - v_B)}{v_A v_B}}{t_R}$$

where $t_R$ is the travel time measured without the anomaly

$$t_R = x_B/v_B$$

With that $S_R$ becomes

$$S_R = \frac{x_A}{x_B}\left(1 - \frac{v_B}{v_A}\right)$$

1070 In the case of a 25% anomaly in the $v$ parameter in P[$\varepsilon$], $v_A$ and $v_B$ are as follows

$$v_A = v_{PA}(1 + \delta \sin^2\theta \cos^2\theta + \varepsilon \sin^4\theta)$$

$$v_B = v_{PB}(1 + \delta \sin^2\theta \cos^2\theta + \varepsilon \sin^4\theta)$$

We use $v_{PA}$ and $v_{PB}$ to refer to the axis-parallel velocity parameter $v$. For a 25% anomaly in $v$ their proportion is

$$v_{PA} = 1.25\, v_{PB}$$

The values for $x_A$ and $x_B$ are

$$x_A = 1\ km \ \underline{and}\ x_B = \ km$$

Thus, the final expression for $S_R$ for a 25% anomaly in $v$ in P[$\varepsilon$] is a constant 0.05 (or 5%).

$$S_R = \frac{1}{5}\left(1 - \frac{1}{1.25}\right) = 0.05$$

1075 Regarding the sensitivity expressed as normalized difference, the general expression for an anomaly in any of the four parameters is

$$S_N = \frac{\dfrac{x_A(v_A - v_B)}{v_A v_B}}{\Delta t_{MAX}} = \frac{x_A}{\Delta t_{MAX}}\left(\frac{1}{v_B} - \frac{1}{v_A}\right)$$

where $\Delta t_{MAX}$ is the maximum travel time difference among all four parameters. This expression will contain some combination of sinusoidal functions for all four possible anomalies.

1080    L198: The authors wrote "..for each of the four parameters. . .25% anomaly was added, while . . . the rest of parameters was homogeneous" – This seems to be contradictory (or it is explained in a misleading way). Parameters v, vT (I use vT instead of the symbol used in the manuscript for v in perpendicular direction) and ε are interrelate by Eq. 2, so changing vT and keeping v constant must change also ε – in result, two parameters are changed. I know that this is not a problem for inversion, as parameters vT and ε are used alternatively, not in the same time, but such description introduces confusion in

1085    interpretation of the sensitivity test results. Also, even if anomalies of every parameter are the same (25%), we can hardly compare cases of ε and vT anomalies: - 25% ε anomaly results in value of 0.2 (compared to background 0.16), while: - for vT, the same anomaly (25%) with respect to background 2.32 results in vT=2.9. This, connected with v in the anomaly being 2.0 (undisturbed, so equal to background), results in ε=0.45 –an anomaly few times larger than in previous case. (This is probably the reason why in Fig. 3 sensitivity for ε reaches ∼0.7%, while sensitivity of vT is few times larger and reaches

1090    4.5%).

There are various points to be addressed in this comment. We most likely did not explain the procedure with sufficient detail. Moreover, we wish to note that this has been a much useful comment that has made us revisit our sensitivity analysis and realize that it needed some corrections. We have modified and reordered sections 3.1 and 3.2 to correctly address these points and to present the new elements in the sensitivity analysis regarding $v$, $\varepsilon$, and $v^{\perp}$. More specifically, we created new

1095    Figures 2 and 3, and included a new Table 1 describing the various model combinations used in both the accuracy and sensitivity analyses.

First of all, just as in the inversion, the forward problem is solved for either one or the other of the two possible parametrizations (P[$\varepsilon$] and P[$v^{\perp}$]), and equation 2 is respected in both cases. When computing the sensitivity for $\varepsilon$ we are

1100    using P[$\varepsilon$] and thus the model is defined by a $v$ model (background), a $\delta$ model (background), and an $\varepsilon$ model (background with the intruded anomaly). Analogously, when calculating the sensitivity for $v^{\perp}$ we are using P[$v^{\perp}$] with a $v$ model (background), a $\delta$ model (background) and a $v^{\perp}$ model (background with intruded anomaly). However, as the referee observed, this combination implies $\varepsilon$=0.45 in P[$\varepsilon$]. This value is greater than the ~0.2 limit assumed for the weak anisotropy approximation [Thomsen, 1986]. Therefore, since it is not possible to establish the comparison between $\varepsilon$ and $v^{\perp}$ based on an

1105    equal anomaly increase of 25% and measuring their respective effects on travel times, we now decided to compare the

sensitivities of these two parameters based on an equal travel time change, i.e. the same change in travel time requires a change of 25% in $\varepsilon$ but only a ~3.4% change in $v^\perp$.

Second, about the comparison of the sensitivities of v, $\delta$, and $\varepsilon$, precisely because we are considering the same relative increase to generate the anomalies, we can most definitely compare the sensitivities for these parameters. We are expressly working with relative changes so as to be able to compare sensitivities for parameters that have significantly different magnitude ranges. We are computing the relative change in travel times produced by proportional perturbations of each of these parameters. In other words, we see how the same proportion of change in each parameter is translated into different relative data changes, i.e. different sensitivities of the data to equal variations in each parameter.

Third, in the case of $v^\perp$, as we just said, we now establish the comparison via the relative increase needed in $\varepsilon$ and $v^\perp$ to achieve the same travel time change. Indeed, in terms of the travel time change produced and following equation 2, for a given v, a certain relative change in $v^\perp$ is equivalent to a greater relative change in $\varepsilon$. This is simply another way of saying that sensitivity to $v^\perp$ is greater than to $\varepsilon$, i.e. to generate the same change in the travel time data, the change in $\varepsilon$ must be of a greater proportion than in $v^\perp$. This is in agreement with the fact that in the inversion tests $v^\perp$ is notably better recovered than $\varepsilon$.

Finally, v is the only parameter in equation 2 that is used in both parametrizations, and so we have to evaluate its sensitivity in both parametrizations. This involves using the background models for $\delta$ and for $\varepsilon$ or $v^\perp$ depending on the case. We made sure that the P[$v^\perp$] case does not violate the weak anisotropy assumption of $\varepsilon$ being smaller or equal to ~0.2 in absolute value, that is, an absolute relative difference between v and $v^\perp$ of up to approximately 20%. The sensitivity (normalized and relative) pattern for v in P[$v^\perp$] is different than for P[$\varepsilon$] although it follows the same sinusoidal pattern and it has equal maxima. Following the mathematical proof in the answer marked with (^) we can see that in the case of P[$v^\perp$] both $S_R$ and $S_N$ will display a sinusoidal shape. Now $v_A$ and $v_B$ are

$$v_A = v_{PA}(1 + \delta \sin^2\theta \cos^2\theta + \left(\frac{v^\perp}{v_{PA}} - 1\right)\sin^4\theta)$$

$$v_B = v_{PB}(1 + \delta \sin^2\theta \cos^2\theta + \left(\frac{v^\perp}{v_{PB}} - 1\right)\sin^4\theta)$$

and the sine and cosine functions do not cancel out in $S_R$, as they did in the case of an anomaly in v in P[$\varepsilon$], nor in $S_N$. For the latter it is trivial to see that the sinusoidal dependencies are identical to the case in P[$\varepsilon$].

As for $\delta$, we computed its sensitivity in both parametrizations and checked that its pattern is independent of the parametrization used as expected. Indeed, in P[$\varepsilon$]

$$v_A = v_P(1 + \delta_A \, sin^2\theta \, cos^2\theta + \varepsilon \, sin^4\theta)$$
$$v_B = v_P(1 + \delta_B \, sin^2\theta \, cos^2\theta + \varepsilon \, sin^4\theta)$$

and in P[$v^\perp$]

$$v_A = v_P(1 + \delta_A \, sin^2\theta \, cos^2\theta + \left(\frac{v^\perp}{v_P} - 1\right) sin^4\theta)$$

$$v_B = v_P(1 + \delta_B \, sin^2\theta \, cos^2\theta + \left(\frac{v^\perp}{v_P} - 1\right) sin^4\theta)$$

and it is trivial to prove that these expressions are equivalent since

$$\varepsilon = \left(\frac{v^\perp}{v_P} - 1\right)$$

-Inversion tests:

The inversion tests are done properly, assuming several variants and various strategies, and results (especially concerning the poor recovery of δ parameter) are interesting, but their main drawback is assumption of near-ideal experimental conditions:

- the measurements geometry (spherical geometry and uniform angular spacing of sources/receivers locations, which results in unrealistically uniform ray coverage).

 -no noise assumed.

Such conditions are almost never possible in case of seismic in-situ experiments, where sources and receivers locations are usually limited to the earth surface, resulting in quite unfavorable ray geometry for solving inversion problem. Therefore, presented tests provide good estimate of 'maximum capabilities' of the code. For the case of modeling the VTI medium in ideal conditions (which is valuable because it shows 'weak points' of the code and parametrization assumed – if the method fails in some aspect in ideal conditions, it will fail even more in case of real data). But such tests give no or very little information about reliability (expected resolution, dependence on the noise level, dependence on the initial model etc.) of a typical seismic experiment.

I think that the manuscript would improve a lot if the authors could add more realistic synthetic tests: in order to properly check behavior of the code in case of typical data form seismic experiment, the tests should assume noisy data, surface location of sources/receivers, and also the dependence of the result on various initial models should be studied.

We are aware that the synthetic experiment that we designed to perform our tests is not realistic. In fact, our goal is to use this canonical benchmark to assess accuracy, sensitivity, and performance of the four inversion strategies under ideal and equal conditions for all anisotropic parameters, avoiding the specificities and biases of a synthetic experiment simulating a particular field case study. These tests on a canonical benchmark provide conclusions that are generally informative of the code's performance. As the referee correctly points out "presented tests provide good estimate of 'maximum capabilities' of the code which is valuable because it shows 'weak points' of the code and parametrization assumed". In other words, here we are not interested in the performance of the code in a more or less specific geological and experimental context, but in obtaining an upper limit to the code's capabilities, an ideal but generalizable estimation for the code's performance.

We have modified the manuscript to clarify the purpose of our testing approach and the reasons behind it. Specifically, we have edited the Abstract (lines 10 and 11), the third paragraph in the Introduction, the first paragraph to section 3 Synthetic tests, and the second paragraph in the Conclusions.

Nonetheless, in an upcoming paper presenting an application of the code to an anisotropic field case, we will first evaluate the code's performance on a realistic synthetic experiment simulating this particular field study. This simulation should yield an estimation of the potential quality of the results as well as information on the best modeling strategy to approach this specific case, and it will include noise, an initial model obtained from isotropic tomography, and will replicate the same exact acquisition geometry used in the field. By synthetically reproducing a more or less specific type of seismic experiment under some realistic circumstances, e.g. a certain noise level, we will get an idea of what to expect in that particular case with the selected noise level, receiver and source densities and distributions, geological features and anomalies, a priori information available in the initial models, etc. Therefore, this sort of realistic synthetic testing and the conclusions that can be drawn from it become relevant and meaningful as preliminary work linked to a particular field data application.

We believe that adding these other type of tests here would result in an exceedingly long manuscript covering too many aspects. Also, we think that realistic synthetic tests are better presented along with the field case that they are simulating. Thus, we prefer to separate our work into two publications, this first one of technical and methodological content, and a second one focusing on the field data application. Figure R1 corresponds to this anisotropic field case. In Sallarès et al. [2013] we obtained an isotropic Vp model from a refraction and wide-angle reflection seismic (WAS) data set (sub-horizontal propagation) and compared it to the image obtained from multichannel seismic (MCS) reflection data (near-vertical propagation). The top plot shows the isotropic Vp model, with the vertical coordinate converted from depth to two-way time (TWT), superimposed onto the MCS image. The white circles inside the model delineate the geometry of inter-plate reflection imaged by MCS data. The thick red line corresponds to the TWT-converted inter-plate boundary obtained from WAS data. The mismatch between the two locations of the inter-plate boundary is most likely due to some degree of seismic anisotropy between near-vertical and sub-horizontal propagations. In the bottom plot we increase the Vp values by

15%, and with this the MCS and WAS locations of the inter-plate boundary now display a good match. Thus, this 15% increase is a good initial estimate of $\varepsilon$, and it indicates, as a general trend, that near-vertical propagation is ~15% faster than sub-horizontal propagation in this area of the subsurface ($\varepsilon \approx 0.15$). For an estimate of $\delta$ we will use the $V_{NMO}$ model from the normal move-out correction of MCS data processing and the isotropic Vp model.

Figs 4-9: It should be marked which column represents which parameter.

The corresponding parameter is indicated within each of the figures in the first row. We have enlarged the four symbols to make them more visible (new Figures 5 to 10).

[Figure]

**Figure R1. The top plot shows the isotropic Vp model, with the vertical coordinate converted from depth to two-way time (TWT), superimposed onto the MCS image. The white circles inside the model delineate the geometry of inter-plate reflection imaged by MCS data. The thick red line corresponds to the TWT-converted inter-plate boundary obtained from WAS data. The mismatch between the two locations of the inter-plate boundary is most likely due to some degree of seismic anisotropy between near-vertical and sub-horizontal propagations. In the bottom plot we increase the Vp values by 15%, and with this the MCS and WAS locations of the inter-plate boundary now display a good match. Ocean bottom receivers are numbered, and the red triangle represents the intersection point with a perpendicular WAS profile.**

[Figure]

**Figure R2: Normalized analytic δ sensitivity for anomalies of 10%, 15%, 20%, and 25%. Sensitivity and anomaly increment**
1205     **display a linear relationship that is particularly clear at the maxima of the curves.**

---

## Author Response (AR2)

[revised manuscript text omitted]

690    In equation (3), second column should be removed from the matrix in present formulation.

Done (see line 145 on page 5).

In Figs 4-9, for clarity, each column should be labelled with name of the parameter represented by given column.

The parameter corresponding to each column is indicated within the panels of the first row at the top left corner. We have

695    enlarged the four symbols to make them more visible (see new Figures 5 to 10).